

# A systematic review of literature on credit card cyber fraud detection using machine and deep learning

Eyad Abdel Latif Marazqah Btoush[1], Xujuan Zhou[1], Raj Gururajan[1,2], Ka Ching Chan[1], Rohan Genrich[1] and Prema Sankaran[3]

[1] School of Business, University of Southern Queensland, Toowoomba, QLD, Australia
[2] School of Computing, SRM Institute of Science and Technology, Chennai, India
[3] School of Management, Presidency University, Bangalore, India

## ABSTRACT

The increasing spread of cyberattacks and crimes makes cyber security a top priority in the banking industry. Credit card cyber fraud is a major security risk worldwide. Conventional anomaly detection and rule-based techniques are two of the most common utilized approaches for detecting cyber fraud, however, they are the most time-consuming, resource-intensive, and inaccurate. Machine learning is one of the techniques gaining popularity and playing a significant role in this field. This study examines and synthesizes previous studies on the credit card cyber fraud detection. This review focuses specifically on exploring machine learning/deep learning approaches. In our review, we identified 181 research articles, published from 2019 to 2021. For the benefit of researchers, review of machine learning/deep learning techniques and their relevance in credit card cyber fraud detection is presented. Our review provides direction for choosing the most suitable techniques. This review also discusses the major problems, gaps, and limits in detecting cyber fraud in credit card and recommend research directions for the future. This comprehensive review enables researchers and banking industry to conduct innovation projects for cyber fraud detection.

## INTRODUCTION

The banking industry has been profoundly impacted by the evolution of information technology (IT). Credit card and online net banking transactions, which are currently the majority of banking system transactions, all present additional vulnerabilities (*Jiang & Broby, 2021*). Hackers have increasingly targeted banks with enormous quantities of client data. Therefore, banks have been in the forefront of cyber security for business. In the past thirteen years, cyber security industry expanded fast. The market is predicted to be valued 170.4 billion in 2022 (*Morgan, 2019*). In the next three years, the cost of cybercrime is expected to rise by 15% every year, finally exceeding $10.5 trillion USD each year by 2025 (*Morgan, 2020*).

Corresponding author
Eyad Abdel Latif Marazqah Btoush,
EyadAbdelLatif.A.Q.
MarazqahBtoush@usq.edu.au

In the banking industry, cyber fraud using credit cards is a significant concern that costs billions of dollars annually. Banking industry has made strengthening cyber security protection a priority. Multiple systems have been developed for monitoring and identifying credit card cyber fraud. However, because of the constantly evolving nature of threats, banking industry must be equipped with the most modern and effective cyber fraud management technologies (*Btoush et al., 2021*).

The acceptance of credit card and other forms of online payments has exploded in recent years, this resulted in an increase in cyber fraud in credit cards. In credit card, there are several forms of cyber fraud. The first type is the actual theft of a credit card. The theft of confidential details of credit card is the second type of cyber fraud. When the credit card information is entered without the cardholder's permission during an online transaction, further fraud is committed (*Al Smadi & Min, 2020*; *Trivedi et al., 2020*).

The detection of cyber fraud in credit cards is a challenging task that attracted the interest of academics working in the fields of machine learning (ML). Datasets associated with credit cards have significant skewness. A great number of algorithms are unable to discriminate items from minority classes when working with datasets that have a considerable skew. In order to achieve efficiency, the systems that are used to identify cyber fraud need to react swiftly. Another important matter of concern is the way in which new methods of attack, influence the conditional distribution of the data over the time period (*Benchaji, Douzi & El Ouahidi, 2021*). According to *Al Rubaie (2021)*, there are a number of challenges need to be addressed for cyber fraud detection in credit card. These challenges contain massive volume of data, that is unbalanced or incorrectly categorised, frequent changes in the type of transaction, and real-time detection.

As current technology being progressed, cyber credit card fraud is also developing rapidly, making cyber fraud detection a crucial area. The conventional techniques to resolve this problem is no longer sufficient. In the conventional technique, domain experts in cyber fraud compose the algorithms which are governed by strict rules. In addition, a proactive strategy must be used to combat cyber fraud. Every industry is attempting to employ ML-based solutions due to their popularity, speed, and effectiveness (*Priya & Saradha, 2021*). ML and DL methods have been shown to be affective in this field. In particular, DL has garnered the most attention and had the most success in combating cyber threats recently. Its ability to minimize overfitting and discover underlying fraud tendencies, as well as its capacity to handle massive datasets, make it particularly useful in this field. In the past few years, DL techniques have been applied to recognize new fraudulent patterns and enable systems to respond flexibly to complex data patterns. In this review, we choose to focus on the latest research from 2019–2021 in order to provide the most up-to-date and relevant information on the topic because DL's popularity has increased during this period.

While there are numerous cyber fraud detection techniques available, as yet no fraud detection systems have been able to deliver high efficiency and high accuracy. Thus it necessary to provide researchers and banking industry with an overview of the state of the art in cyber fraud detection and an analysis of the most recent studies in this field to conduct innovation projects for cyber fraud detection. To achieve this goal, this review will

provide a detailed analysis of ML/DL techniques and their function in credit card cyber fraud detection and also offer recommendations for selecting the most suitable techniques for detecting cyber fraud. The study also includes the trends of research, gaps, future direction, and limitations in detecting cyber fraud in credit card.

This review focuses mostly on identifying the ML/DL techniques used to detect credit card cyber fraud. Moreover, we aim to analyse the gaps and trends in this field. Over the past few years, there have only been a few review articles published on detecting credit card cyber fraud. This review takes a look at the detection of card fraud from the standpoint of cybersecurity and applies ML/DL techniques and approached the topic from a financial standpoint. Furthermore, unlike other reviews, which also include conference article, ours only includes recent journal articles.

The aim of this review is to provide researchers with an overview of the state of the art in cyber fraud detection and an analysis of the most recent studies in this field. This review will assist researchers in selecting high-performance ML/DL algorithms and datasets to consider when attempting to detect cyber fraud. To answer the four research questions, we have utilized the search string to conduct research in six digital libraries. This resulted in a total of 2,094 article, all of which are journal article. In addition, we utilised the snowballing strategy to integrate more relevant articles missed by the automated search. Through careful referencing of the explored article, we have narrow down our collection and found the most relevant answers for our four research questions. As a result, 181 article were chosen for further study.

We describe our search study selection, data extraction procedures, and overall research methodology in "Survey Methodology" of this article. In "Result and Analysis", we present the findings and answers to our research questions. In "Conclusions", we conclude the study by discussing its findings.

## SURVEY METHODOLOGY

The review investigates the present status of research on detecting cyber fraud in credit card and addresses our research questions. The methodology begins with a description of the data sources, the search strategy, the inclusion and exclusion criteria, as well as the quantity of research article selected from the different databases.

### Research questions

This review attempts to summarise and analyse the ML and DL credit card cyber fraud detection algorithms from 2019 to 2021. The following research questions (RQs) are therefore posed:

**RQ1:** What ML/DL techniques are utilised in detection of credit card cyber fraud? This question aims to specify the ML/DL techniques that have been applied.

**RQ2:** What percentage of credit card cyber fraud detection articles discussed supervised, unsupervised, or semi-supervised techniques? This question seeks to determine the proportion of research articles that employ supervised, unsupervised, and semi-supervised credit cyber fraud detection techniques.

**RQ3:** What is the estimated overall performance and outcomes of ML/DL models? This question focuses on ML/DL model performance estimation and model results.

**RQ4:** What are the research trends, gaps, and potential future directions for cyber fraud detection in credit card? The question guides to uncover research trends, gaps in the existing literature, and future direction of credit card cyber fraud research.

## Data sources and research strategy

After determining the research questions, we constructed the research as follows:

– The main search terms are determined by the research questions.
– Boolean operators (AND and OR) are used to restrict search results.
– The search terms utilised for this review are related to detect cyber fraud in credit card and ML/DL techniques used for fraud detection.

The methodology incorporates the following electronic literature databases in order to obtain a comprehensive and broad coverage of the literature and to maximise the probability of discovering highly relevant articles:

– Google Scholar—ACM—IEEE Xplore—SpringerLink—Web of Science—Scopus.

For the purpose of locating the most relevant article, particular Keywords were formulated into a search string. This string was divided into search units and Boolean operators were used to combine them. All of the mentioned resources have keyword-based search engines. We selected the following search string to retrieve the most relevant studies:

| Box 1 |
| --- |
| ((AI OR "artificial intelligence" OR DL OR "deep learning" OR ML OR "machine learning") AND ("Credit card fraud" OR "card fraud" OR "card-fraud" OR "credit-fraud" OR "card cyber fraud" OR "transaction fraud" OR "payment fraud" OR "fraud detec*" OR "bank* fraud" OR "financ* fraud")). |

We include "artificial intelligence" OR "deep learning" OR "machine learning" thus that we can find studies that utilised any of these techniques. Additionally, we included the "credit card fraud" OR "card fraud" OR "card-fraud" OR "credit-fraud" OR "card cyber fraud" OR "transaction fraud" OR "payment fraud" OR "fraud detec*" OR "bank* fraud" OR "financ* fraud" term to concentrate on any fraud-related content so that we do not miss any relevant articles.

We conducted a search for the above string in six digital libraries. The research string is edited and converted into an appropriate search query input for each library. Table 1 provides the detailed search queries. We limited our review to journal articles, excluding conference article, books, and other publications. In December 2021, our search conducted for the years from 2019 to 2021. There were a total of 2,094 items retrieved from research libraries. Table 2 depicts the distribution of the items throughout the libraries. We identified 365 duplicate article. After eliminating the duplicates, we continued with the

**Table 1 The search queries in detail for six digital libraries.**

| Digital library | Query |
|---|---|
| Google scholar | ((AI OR "artificial intelligence" OR DL OR "deep learning" OR ML OR "machine learning") AND ("Credit card fraud" OR "card fraud" OR "card-fraud" OR "credit-fraud" OR "card cyber fraud" OR "transaction fraud" OR "payment fraud" OR "fraud detec*" OR "bank* fraud" OR "financ* fraud")). |
| ACM | ((All: AI) OR (All: "artificial intelligence") OR (All: DL) OR (All: "deep learning") OR (All: ML) OR (All: "machine learning")) AND ((All: "credit card fraud") OR (All: "card fraud") OR (All: "card-fraud") OR (All: "credit-fraud") OR (All: "card cyber fraud") OR (All: "transaction fraud") OR (All: "payment fraud") OR (All: "fraud detec*") OR (All: "bank* fraud") OR (All: "financ* fraud")) AND (Publication date: (01/01/2019 TO 12/31/2021)) |
| IEEE Xplore | ((AI OR "artificial intelligence" OR DL OR "deep learning" OR ML OR "machine learning") AND ("Credit card fraud" OR "card fraud" OR "card-fraud" OR "credit-fraud" OR "card cyber fraud" OR "transaction fraud" OR "payment fraud" OR "fraud detec*" OR "bank* fraud" OR "financ* fraud")). Filters applied: Journals 2019–2021. |
| Springerlink | 39 Result(s) for '((AI OR "artificial intelligence" OR DL OR "deep learning" OR ML OR "machine learning") AND ("Credit card fraud" OR "card fraud" OR "card-fraud" OR "credit-fraud" OR "card cyber fraud" OR "transaction fraud" OR "payment fraud" OR "fraud detec*" OR "bank* fraud" OR "financ* fraud"))' within article 2019–2021. |
| Web of science | ((AI OR "artificial intelligence" OR DL OR "deep learning" OR ML OR "machine learning") AND ("Credit card fraud" OR "card fraud" OR "card-fraud" OR "credit-fraud" OR "card cyber fraud" OR "transaction fraud" OR "payment fraud" OR "fraud detec*" OR "bank* fraud" OR "financ* fraud")). Refined by: publication years: 2019 or 2020 or 2021 Document types: Articles languages: English. |
| Scopus | TITLE-ABS-KEY (((AI OR "artificial intelligence" OR DL OR "deep learning" OR ML OR "machine learning") AND ("Credit card fraud" OR "card fraud" OR "card-fraud" OR "credit-fraud" OR "card cyber fraud" OR "transaction fraud" OR "payment fraud" OR "fraud detec*" OR "bank* fraud" OR "financ* fraud"))) AND (LIMIT-TO (PUBYEAR, 2021) OR LIMIT-TO (PUBYEAR, 2020) OR LIMIT-TO (PUBYEAR, 2019)) AND (LIMIT-TO (DOCTYPE, "AR")). |

**Table 2 Count of article collected from the six libraries.**

| NO | Database | Web address | Retrieved article |
|---|---|---|---|
| 1 | Google scholar | https://scholar.google.com/ | 1,418 |
| 2 | Springerlink | https://link.springer.com/ | 39 |
| 3 | Scopus | https://www.scopus.com/ | 292 |
| 4 | IEEE Xplore | https://ieeexplore.ieee.org | 76 |
| 5 | Web of science | https://webofknowledge.com/ | 233 |
| 6 | ACM | https://dl.acm.org/ | 36 |
| Total of retrieved article 2,094 | | | |
| The number of duplicates 365 | | | |
| The number of article after removing duplicates 1,729 | | | |

selection process based on the remaining 1,729 article. In addition to the automatic searches of digital libraries, snowballing mechanism was also used.

## Study selection

We executed the above search strategy during December 2021 and identified 2,094 article. After removing duplicates (365 articles), the titles and abstracts of 1,729 unique citations were screened for eligibility. We screened the titles and abstracts for relevance. If the study's relevance could not be verified due to insufficient abstract information or the

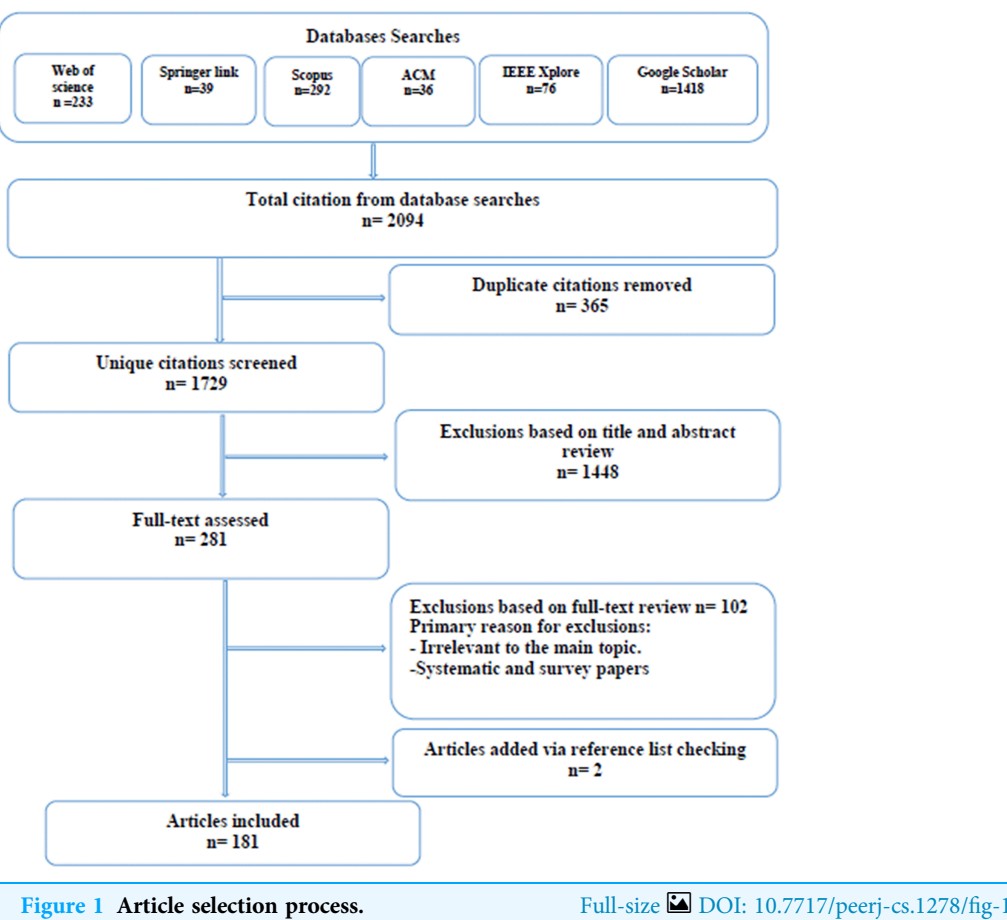

**Figure 1 Article selection process.**

absence of an abstract, the citation was assigned for full-text review. Thus we reviewed the full text of 281 studies. Disagreements on the included studies were resolved through discussion and consensus. The selected article were filtered to ensure that only relevant studies were included in our review. Then the article were exported to EndNote and grouped for each database and then exported to a literature review management software called Rayyan (*Ouzzani et al., 2016*) to facilitate the screening and selection process. To initiate the filtering and selection processes, duplicate articles gathered from multiple digital resources are eliminated. Then using inclusion and exclusion criteria, removed the irrelevant article. Using quality evaluation processes we included only the qualified article that offer the most effective answers to our study objectives. Using the collected article references, we searched for further related publications. Figure 1 displays the article selection process. The inclusion and exclusion criteria utilised for this review are detailed in Table 3. After the filtration process was completed, 181 article were observed for this study.

## Data extraction

This process aims to analyse the final selection of article in order to collect the data required to answer the four research questions. Table 4 displays our data extraction form. In the final column of Table 4, the reason for extracting the corresponding data were given.

**Table 3  Criteria of inclusion and exclusion.**

| Inclusion criteria | Exclusion criteria |
| --- | --- |
| Include journal article only | Exclude conference article, chapter book, and other publication. |
| Include articles about credit card cyber fraud detection | Exclude articles not related to detect cyber fraud in credit card |
| Include articles that used ML/DL | Exclude articles that did not use ML/DL |
| Include articles published in 2019, 2020, and 2021 | Exclude articles that published before 2019 and after 2021 |
| Include articles in English language | Exclude publications in languages other than English. |

**Table 4  Extraction form.**

| Strategy | Category | Description | Purpose |
| --- | --- | --- | --- |
| Automatic extraction | Title of article | the article's title | Additional information |
| | Authors of article | The author's name | |
| | Article year | The year of publication | |
| | Article type | Journal | |
| Manual extraction | Objectives | study objectives | RQ4 |
| | Conclusion | Outcomes of study | RQ4 |
| | Techniques | ML/DL technique utilised to support objectives | RQ1 and RQ2 |
| | Discussion and result | Outcomes | RQ3 |
| | Algorithm type | ML, DL, or mix | RQ1 and RQ2 |
| | Dataset | Dataset used in article | RQ1 |
| | Future work | Gaps, trends, and future work | RQ4 |

We answered RQ1 and RQ2 using information regarding techniques and datasets. We used this information to group studies with comparable datasets and techniques. Extraction of each article's discussion and findings was an aid in estimating the overall performance of approaches and answering RQ3. By extracting out the article' objectives and conclusions, we are able to recognise trends, conduct gap analysis, determine future research, and provide a response to RQ4. As a result, in order to identify the gaps and define the next direction of future research should take, on the basis of the article's objectives and conclusions, we conducted a summary analysis.

## RESULT AND ANALYSIS

### Distribution of chosen articles throughout the years

To explore the most recent techniques described in journals published in this field, limits were placed on publishing years. Our review selected article that were published from 2019 to 2021. In Fig. 2 we specified the distribution of article by year of publication. Since our study was completed in December 2021, it is important to note that article published after December 2021 were not included.

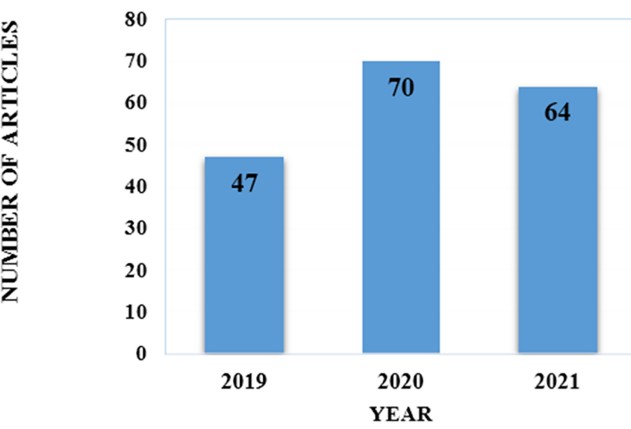

**Figure 2 Distribution of the selected articles over the publication year.**

## Publication type

In this review, we evaluated only journal publications. Table A1 displays the selected research articles published during the observation period.

## Data synthesis results

This section examines the ultimately selected article (181 article). In order to provide a response to each of our four research questions, a synthesis of the data is performed. For RQ1: What types of ML/DL algorithms and datasets are used in credit card cyber fraud detection?

## Cyber fraud detection techniques

In this part we address RQ1, which seeks to specify the ML/DL techniques used in detecting cyber fraud in credit card from 2019 to 2021.

### *Machine learning*

ML identified as a technique relevant to a wide range of problems, especially in sectors requiring data analysis and processing. ML, which is classified as supervised ML, unsupervised ML, and reinforced ML, plays a crucial role in resolving the unbalanced dataset. ML techniques are tremendously effective for detecting and preventing fraud because they enable the automated recognition of patterns across vast amounts of data. Adopting the proper ML models facilitates the differentiation between fraudulent and legitimate behaviour. These clever systems may adapt over time to new, unseen fraud schemes. Thousands of computations must be executed correctly in milliseconds for this to be possible. Both supervised and unsupervised technologies help detect cyber fraud and must be included in the future generation of fraud safeguards.

Supervised Learning is the training technique for ML algorithms on labelled data sets and configurable data with known variable targets. Classification, regression, and inference are all instances of supervised learning. In all field, supervised models that are trained on a large number of accurately labelled transactions are the most common ML technique. Each transaction is classified as either fraudulent or legitimate. The models are trained by giving

them voluminous labelled transaction data in order for them to discover patterns that best resemble genuine behaviour.

Unsupervised learning is the process of training a ML algorithm on a dataset containing ambiguous target variables. The model make an effort to discover the most significant patterns in data. Unsupervised learning technique include dimension removal and cluster segmentation.

Semi-supervised learning combines supervised and unsupervised learning by training model on unlabeled data. In this method, the unsupervised learning attribute is utilised to determine the optimal data representation, while the directed learning attribute is used to analyse the relationships within that representation and subsequently create predictions.

Multiple research utilised supervised, unsupervised, and semi-supervised ML approaches. Table B1 displays the frequency of use of ML and DL techniques in the reviewed literature, indicating how often each technique type is utilised. Several article utilised several ML/DL techniques, as should be highlighted.

*Supervised techniques*
**Classification techniques**

Utilizing supervised algorithms is the most common method for detecting credit card cyber fraud. Various supervised models are utilised in this field. Support vector machine (SVM) utilised to classify data samples into two groups using a maximum margin hyper plane. It specifically classifies fresh data points using a labelled dataset for every category. The SVM used in 56 reviewed articles. SVM's kernel consists of mathematical functions that convert input data to high-dimensional space. Therefore, SVM can classify linear and nonlinear (using kernel function) data.

Linear, radial, polynomial, and sigmoid are the four types of kernel functions, utilised in *Li et al. (2021)*, this article uses SVM to detect credit card fraud. Using cuckoo search algorithm (CS) and genetic algorithm (GA) with particle swarm optimisation technique to optimise the SVM parameters (PSO). Experiments have shown that the linear kernel function is the most effective function. Kernel function is optimised using radial basis function. In terms of overall performance, PSO-SVM outperforms CS-SVM and GA-SVM.

*Pavithra & Thangadurai (2019)* suggested a hybrid architecture involving the optimization of the particles swarm (PSO). Feature selection algorithm based on SVM was used to improve prediction of cyber fraud. Results shown PSO-SVM method is an optimal preparatory instrument for enhancing feature selection optimisation. In *Zhang, Bhandari & Black (2020)*, a weighted SVM algorithm is utilised. Experiments revealed that this model significantly enhance the performance. Weighted feature based SVM (WFSVM) with time varying inertia weight base dragonfly algorithm (TVIWDA) proposed in *Arun & Venkatachalapathy (2021)*. TVIWDA-optimized property is chosen to increase the detection accuracy. Then, using the WFSVM classifier and the specified characteristics, the classification is performed. The results shown that the suggested model outperforms the current random tree based technique. WFSVM is more efficient with smaller datasets.

The decision tree (DT) approach has gained remarkable interest from researchers. The DT algorithm appeared in 49 articles. In *Bandyopadhyay et al. (2021)*, the DT classifier

applied for detection of financial frauds. DT algorithm performs the best with an accuracy of (0.99) comparing with another classifier. DT with boosting technique applied in *Barahim et al. (2019)*. The results show that applying boosting with DT outperforms other methods. The model obtained highest accuracy of 98.3%. In *Choubey & Gautam (2020)*, a combination of supervised algorithms such as DT, RF, LR, naive Bayes (NB), and K-near neighbor (KNN) have been utilised. The study observed that hybrid classifier DT with KNN worked better than any other single classifier. In *Hammed & Soyemi (2020)*, the utilisation of the DT algorithm enhanced with regression analysis is described. The result indicates enhanced performance. This approach is accurate, with a misclassification error rate of 18.4%, and the system successfully validated all of the inserted incursions used for testing.

Among ML approaches, the C4.5 algorithm acts a DT classifier. The decision is based on certain occurrences of data. Four articles utilised C4.5 tree (*Askari & Hussain, 2020*; *Beigi & Amin Naseri, 2020*; *Husejinovic, 2020*; *Mijwil & Salem, 2020*). New model applied C4.5 in *Mijwil & Salem (2020)*. The study revealed that C4.5 is the best classifier comparing with other ML techniques. Credit card fraud detection using C4.5 DT classifier with bagging ensemble has been applied in *Husejinovic (2020)*. The study revealed that bagging with C4.5 DT is the best algorithm. Logistic model tree (LMT) has been used in DT for classification. In *Hussein, Abbas & Mahdi (2021)*, LMT applied to fraud classification and detection. The result shows that applying LMT algorithm to classification fraud is better than other techniques. LMT model obtained 82.08% accuracy. Intuitionistic fuzzy logic based DT (IFDTC4. 5) applied in *Askari & Hussain (2020)* for transaction fraud detection. The results show that the IFDTC4.5 outperforms other techniques and able to detect fraud proficiently.

One of the most powerful techniques is RF, which is a modern variation of DT. According to the examined literature, RF is the most prevalent credit card fraud detection method (74 articles). Some reviewed articles used RF only for comparison with the developed methods. In *Amusan et al. (2021)*, RF applied for fraud detecting on skewed data. Results indicated that RF recorded highest accuracy (95.19%) comparing with KNN, LR, and DT. Furthermore, RF applied with other techniques such as SVM, NB, and KNN in *Ata & Hazim (2020)*. The results showed that RF algorithm performs better than the other techniques. A hybrid model or combination of supervised classifiers appeared in *Choubey & Gautam (2020)*. Several techniques such as RF, KNN, and LR have been applied. Results show that RF with KNN worked better than applied as a single classifier.

New model applied RF in *Meenakshi et al. (2019)*. The study revealed that the RF algorithm performs better with more training data, but testing and application speeds will decrease. *Jonnalagadda, Gupta & Sen (2019)* applied RF in their study. The recommended values for the highest level of RF precision are 98.6%. This proposed module is suitable to a larger data set and yields more precise results. With more training data, RF algorithm will perform better. In *Hema & Muttipati (2020)* LR, RF, and Catboost have been applied for discovering cyber fraud. The result shows RF with Catboost gives high accuracy. RF gives the best result with accuracy (99.95). RF with SMOTE applied in *Ahirwar, Sharma & Bano (2020)*. The results obtained by the RF algorithm showed that this approach would be

successful in real time. This model is intended to have some insight into the identification of fraud.

Bayesian technique is an additional classification method. We explored 42 articles that utilised NB, and two articles used Bayesian belief networks (BBN). Detection of credit card fraud *via* NB and robust scaling approaches described in *Borse, Patil & Dhotre (2021)*. The results indicate that the NB classifier with the robust scaleris is the most effective in predicting fraudulent activity in the dataset. NB using robust scaling got the accuracy 97.78%. In *Divakar & Chitharanjan (2019)*, the NB classifier and other classifiers were applied. NB did not obtain the best result when comparing with other classifiers. In *Gupta, Lohani & Manchanda (2021)*, among ML algorithms such as LR, RF, and SVM, the NB algorithm's performance is remarkable. BBN applied in *Kumar, Mubarak & Dhanush (2020)* for detecting fraud in credit card. Result showed a BBN is more accurate than the NB classifier. This is disturbed with using the fact of conditional dependence between the attributes in Bayesian network, but it requires more calculation and training process. The transaction of data value available in dataset which is trained with their results as fraud or genuine transaction which is predicted by a testing data value for individual transaction.

The K-nearest neighbors (KNN) algorithm applied in 39 articles. Various studies were used KNN technique in detecting credit card fraud. KNN uses neighbouring samples to identify class label. The KNN technique is best for overlapping sample sets (*Yao et al., 2019*). In this review, several articles applied KNN as classifiers. *Chowdari (2021)* reported that the KNN is a stronger classifier at detecting fraud in credit cards comparing with other techniques such as DT, LR, and RF. In *DeepaShree et al. (2019)*, *Kumar, Student & Budihul (2020)*, the KNN classifier applied for credit card fraudulent transaction detection, comparing with RF and NB, KNN showed the highest accuracy than the RF algorithm and NB. In *Parmar, Patel & Savsani (2020)* and *Vengatesan et al. (2020)*, the KNN technique compared with many other techniques such SVM, LR, DT, RF XGBoost. The KNN model is the most precise model. KNN model got accuracy score: 99.95%. New ML approach to detect anonymous fraud patterns appeared in *Manlangit, Azam & Shanmugam (2019)*, Synthetic minority oversampling technique (SMOTE) with KNN proposed. Results reveal that proposed model performed well. KNN model achieves a precision 98.32% and 97.44.

**Regression techniques**

In this review, the studies utilised logistic regression (LR) technique frequently. A total of 52 studies employed LR for cyber fraud detection. LR models can be utilised for both multiclass and binary classification. LR is a statistical strategy that models a binary dependent variable using a logistic function. In *Adityasundar et al. (2020)*, LR applied over highly imbalanced data. Using unbalanced data, the study developed a classification model that is extremely resistant. New system uses LR to build the classifier proposed in *Alenzi & Aljehane (2020)*. Comparing the proposed LR-based classifier against the KNN and voting classifiers. The result demonstrates that LR-based produces the most accurate findings, with a 97.2% success. *Itoo & Singh (2021)* revealed a comparison between LR, NB, and KNN for fraud detection. Results show that LR achieved an optimal performance. LR was successful in achieving greater accuracy than KNN and NB. The LR attained accuracy of 95%, while the NB achieved 91%, and the KNN achieved 75% (*Itoo & Singh, 2021*). In

*Karthik et al. (2019)*, a newly proposed approach shown that employing a stacking classifier that applies LR as a meta classifier is the most promising method, followed by SVM, KNN, and LR. A study by *Soh & Yusuf (2019)* suggested four models to detect fraud on an imbalanced data. Result shows that the RF and KNN are overfitting. Thus, only the DT and LR have been compared. The result shows that LR with stepwise splitting rules has outperformed the DT with only 0.6% error rate. *Sujatha (2019)* used single and hybrid model of under sampling and over sampling. The study revealed that LR is best among all the algorithms. The result shows that the proposed model LR and NN approaches outperform DT.

**Ensemble techniques**

Random forest model is an ensemble approach appeared in the examined literature. RF often achieves superior performance against single DT by producing a stack of DT over training. New research conducted in 2021 revealed that RF outperforms K-means and SVM (*Al Rubaie, 2021*).

Another ensemble method is bagging, which is a collection of different estimators created using a particular learning process to enhance a single estimator. Bagging reduces DT classifier variance. The approach creates random subsets from the training sample. In the reviewed articles, five article applied bagging methods (*Alias, Ibrahim & Zin, 2019*; *Husejinovic, 2020*; *Lin & Jiang, 2021*; *Mijwil & Salem, 2020*; *Karthik, Mishra & Reddy, 2022*). *Husejinovic (2020)* applied C4.5 DT, NB, and bagging ensemble to predict fraud. Result shows that best algorithm is bagging with C4.5 DT.

Boosting includes adaptive boosting algorithm (AdaBoost), RUSBoost, gradient boosting algorithm (GBM), LightGBM, and XG Boost algorithm. A total of 59 articles utilised boosting techniques in the reviewed articles. AdaBoost employed by *Barahim et al. (2019)*. In this study, DT, NB, and SVM used with AdaBoost. The results show that AdaBoost with DT outperforms other techniques. A comparison of different ensemble methods to predict fraud in credit card has been done by *Faraj, Mahmud & Rashid (2021)*. Experiment shows that XGBoosting performs better when compared to other ensemble methods and also better than neural networks.

Stacking is a method of ensemble learning that combines multiple classification or regression systems. In stacking, a single model used to exactly integrate predictions from contributing models, but in boosting, a series of models are utilised to enhance the predictions of earlier models. In contrast to bagging, utilising the complete data set as compared to portions of the training dataset. Four articles have been used stacking to learn a classifier for detecting fraud in credit card (*Karthik et al., 2019*; *Muaz, Jayabalan & Thiruchelvam, 2020*; *Prabhakara et al., 2019*; *Veigas, Regulagadda & Kokatnoor, 2021*). The stacked ensemble approach has demonstrated potential for detecting fraudulent transactions. Stacked ensemble has the best performance at 0.78 after trained for sampled datasets (*Muaz, Jayabalan & Thiruchelvam, 2020*).

*Unsupervised techniques*

Clustering is the process of categorising similar instances into identical groupings. The clustering methods utilised far less comparing with classification methods in the reviewed

article. The hidden Markov model is used to model probability distribution across sequences of observation. It consists of hidden states and observable outputs. HMM has been applied in seven articles. In *Das et al. (2020)*, HMM model applied to detect cyber. Results show a great performance of proposed system, also demonstrate advantage of learning cardholder's spending behaviour. *Singh et al. (2019)* suggested method to identify cardholders spending profile, then attempts to find out the observation symbols, these observation symbols will help for an initial estimate of the model parameters. Thus, HMM can detect if the transaction is genuine or fraud. SMOTE utilised along with HMM and density based spatial clustering of application and noise. This new model (SMOTE +DBSCAN+HMM) performed relatively better for all the various hidden states.

K-means has been applied in seven articles. The K-means algorithm is a non-hierarchical method applied for data clustering. The algorithm uses a simple method. Thus, K-means classifies a given dataset into a specified number of clusters or K-clusters. In *Abdulsalami et al. (2019)*, K-mean was applied with back-propagation neural network (BPNN). The result shows that there is a significance difference between BPNN and K-means for detecting fraud credit card transaction. The BPNN model achieved a great accuracy with less false alarms comparing with K-means model. Results also show that the accuracy of BPNN is 93.1% while K-means accuracy is 79.9%.

Isolation forest is an unsupervised ensemble. No point-based distance calculation and no profiling of regular instances are done. Instead, the Isolation forest builds an ensemble of DTs. The concept of isolation forest is to spilt anomalies with the purpose of isolation them. An ensemble of DTs is generated for a particular data collection, the data points with the shortest average path length are considered anomalous. Isolation forest has been applied in 19 articles. In *Meenu et al. (2020)*, a new Isolation Forest model to detect fraud is utilised. The model demonstrates the efficiency in fraud detection, observed to be 98.72%, which indicates a significantly better approach than other fraud detection techniques. Isolation forest with local outlier factor to detect fraud applied in *Vijayakumar et al. (2020)*. Isolation forest showed accuracy as 99.72% while local outlier factor showed accuracy as 99.62%. Isolation factor is better observed in online transactions. A study by *Palekar et al. (2020)* that K-means clustering and (Isolation forest and local outlier factor) can be created and developed on a very large scale to detect fraud in credit card transaction.

Self-organising map (SOM) is unsupervised neural networks learning (NN). SOM is appropriate for building and analysing the profiles of customers to detect fraud. SOM applied in two reviewed articles. SOM and NN in hybrid approach applied in *Harwani et al. (2020)*. Compared to using SOM and ANN alone, the suggested model reached a better accuracy and cost. In *Deb, Ghosal & Bose (2021)*, three unsupervised algorithms, K-means, K-means clustering using principle component analysis (PCA), T-distributed stochastic neighbor embedding (T-SNE), and SOM are presented. This model achieved accuracy of 90% for fraud detection in credit card. The results show also K-means clustering along with PCA is much better than simple K-means. Also, T-SNE is much better than PCA as the PCA gets highly affected by outliers.

*Semi-supervised techniques*

A hybrid technique combining supervised and unsupervised learning. The unsupervised learning attribute is utilised to determine the optimal representation of data, whereas the supervised learning attribute is employed to investigate the relationships in the representation before beginning to predict. Semi-supervised learning is extremely useful when the data collection is unbalanced. The studies in this review utilised semi-supervised technique in their researches. Three studies employed semi-supervised to detect fraud in credit card (*Dzakiyullah, Pramuntadi & Fauziyyah, 2021*; *Pratap & Vijayaraghavulu, 2021*; *Shekar & Ramakrisha, 2021*). In *Dzakiyullah, Pramuntadi & Fauziyyah (2021)*, a combination of semi-supervised learning and AutoEncoders to detect fraudulent transaction is presented. This proposed model utilized an autoencoder then trains the basic linear classifier to allocate the data collection into own class. Also, the T-SNE applied to visualise the essence of fraudulent and non-fraudulent transactions. Results obtained are helpful because that credit card fraud will be easily classified with 0.98%.

Semi supervised algorithms using majority voting applied in *Pratap & Vijayaraghavulu (2021)*; in this study, 12 ML algorithms applied. Firstly, the standard models are used. Secondly, AdaBoost and majority voting added. Result indicates that the Majority voting technique achieves high accuracy.

### Deep learning

Deep learning (DL) is subsection of ML uses data to teach computers how to perform tasks. The fundamental tenet of DL is that as we expand our NN and train them with new data, their performance continues to improve. The main advantage of DL over traditional ML is its higher performance on large datasets. The most frequently used DL algorithms in cybersecurity are feed forwards neural networks (FNNs), stacked autoencoders (SAE), and convolutional neural networks (CNNs). As shown in Fig. 3, DL techniques have been used in 34 reviewed articles. A total of 39 reviewed articles used combination of DL and ML techniques to detect fraud in credit card.

An artificial neural network (ANN) employs cognitive computing to aid in the development of machines capable of employing self-learning algorithms including pattern recognition, natural language processing, and data mining. ANN presents more accurate results because it learns from the patterns of authorized behaviour and thus distinguishes between 'fraud' and 'non-fraud' in credit card transaction. We explored 36 articles that used ANN in our review. In *Agarwal (2021)*, ANN implemented for identity theft detection. The proposed model aims to use the different layers in a NN to determine the fraud transaction. The result shows that applying an ANN gives accuracy nearly equal to 100%. The result shows that ANN is best suited for determining if a transaction is fraudulent or not. New recent study applied ANN to detect fraud. The ANN technique has been used then compared with ML algorithms such as SVM, KNN. The result shows that ANN gives accuracy more than other ML algorithms, the suggested model is optimal for detecting credit card fraud (*Asha & Suresh Kumar, 2021*).

In *Abdulsalami et al. (2019)*, back-propagation neural network (BPNN) and K-means are applied. The results indicate that the BPNN is more accurate than K-means algorithm.

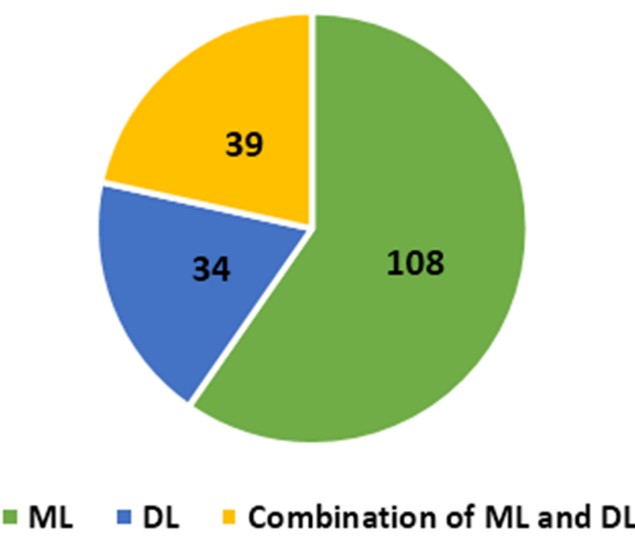

**Figure 3** **Number of articles applied ML/DL techniques.**

BPNN obtained accuracy of 79.9%. The results also indicate that K-means reduced prediction time provided it and advantage over BPNN. In *Daliri (2020)* harmony search algorithm with ANN (NNHS) are applied to improve fraud detection in banking system. The results show acceptable capability in fraud detection based on the information of customers. In *Oumar & Augustin (2019)* ANN with LR applied for fraud detection. Back-propagation has decreased the error function and enabled the model to discriminate between a fraudulent and a legitimate transaction. The suggested model is 99.48% accurate in its predictions and highly reliable.

Multilayer perceptron (MLP) is the most approach in ML because to its excellent accuracy in approximation nonlinear function. MLP comprises of three distinct layers. We explored 14 articles that used MLP in our review. In *Alias, Ibrahim & Zin (2019)*, MLP and fifteen other types of supervised ML techniques are examined to determine the one with highest accuracy for detecting fraudulent transaction. The result shows that MLP generated the greatest detection accuracy of 15 algorithms, at 98%. *Can et al. (2020)* applied MLP and other ML techniques such as DT, RF, and NB. Regarding amount-based profiling, both MLP and classifiers demonstrated substantial improvements. In *Faridpour & Moradi (2020)*, a novel ML-based model for detecting fraud in banking transaction utilising customer profile data is provided. In the proposed model, bank transactional data is utilised and an MLP with adjustable learning rate is trained to demonstrate the transaction authenticity, thus improving detection process. The suggested model surpasses SVM and LR. The accuracy of the proposed model is 0.9990.

Convolution neural network (CNN) is composed of multiple layers, output of which are used as inputs to layers that follow. ConvNET's purpose is to reduce the input into a framework that is easier to comprehend, without sacrificing crucial information for making accurate predictions. CNN used in seven articles in the review. In *Agarwal et al. (2021)*, DL techniques like CNN, BILSTM with ATTENTION layer have been used to

detect and classify the illegitimate transactions. The CNN-Bi-LSTM-ATTENTION model detects the fraudulent class with high accuracy. Analysis shows that the model is adequate and yields an accuracy of 95%. The results demonstrate that the addition of an attention layer increases the performance of the model, allowing it to accurately discriminate between fraudulent and legitimate transactions. A CNN, NB, DT, and RF hybrid model is deployed in *Aswathy & Samuel (2019)*, these algorithms are used as single models. Then these are used as hybrid models using majority voting technique. Adaptive boosting algorithm was used to boost the performance of classifiers.

DNNs, which provide potent tools for automatically producing high-level abstractions of complicated multimodal data, have recently garnered a great deal of interest from business and academics. DNNs learn features on their own, resulting in an increasingly accurate learning process. DNNs have been shown to be more efficient and accurate. Four studies employed DNN. In *Arya & Sastry (2020)*, the proposed model is flexible to data disparity and resistant to hidden transaction patterns. Adaptive optimisation is recommended to improve fraud prediction. Result demonstrates its superiority over current other methods.

Credit card fraud detection using uncertainty-aware DL was implemented in *Habibpour et al. (2021)*. It is vital to evaluate the uncertainty of DNN predictions. According to the study, there are three uncertainty quantification (UQ) techniques, ensemble, Monte Carlo dropout, and ensemble Monte Carlo dropout that can be used to quantify the level of uncertainty associated with predictions and produce a categorisation that is reliable. According to the findings, the ensemble method is superior at capturing the uncertainty related to predictions.

Deep convolution neural network (DCNN) applied in four articles. The DCNN technique can improve detection accuracy when a huge volume of data is involved. In *Chen & Lai (2021)*, existing ML models, including LR, SVM, and RF, as well as auto-encoder and other DL models. Results show a detection accuracy of 99% was attained over a 45-s duration. Despite the vast quantity of data, the model provides enhanced detection. DL technique provides high accuracy and rapid pattern in detecting complex and unknown patterns. 1DCNN, 2DCNN, and DCNN have also been utilised to detect credit card cyber fraud in *Cheng et al. (2020)*, *Deepika & Senthil (2019)*, *Nguyen et al. (2020)*.

A recurrent neural network, often known as an RNN, is a structure that used to remember previous input sequences. It is comprised of links between the internal nodes of a directed graph. Depending on the amount of their internal memory. RNN applied in seven articles in this review (*Bandyopadhyay & Dutta, 2020*; *Chen & Lai, 2021*; *Forough & Momtazi, 2021*; *Hussein et al., 2021*; *Osegi & Jumbo, 2021*; *Sadgali, Sael & Benabbou, 2021*; *Zhang et al., 2021*). In *Bandyopadhyay & Dutta (2020)*, Implementing and applying RNN on synthetic dataset. The suggested model can detect fraudulent transaction with a 99.87% accuracy. The outcomes demonstrate that the approach is relevant and appropriate for detecting fraud. In *Forough & Momtazi (2021)*, a deep RNN-based ensemble model and an ANN-based voting approach proposed. The ensemble model leverages a variety of RNN as the fundamental classifier and combines output using an FFNN as voting method. Classification employs a number of GRU or LSTM network. The outcomes indicate that

the suggested model outperforms competing models. The proposed model is superior to existing models in this field. Bidirectional gated recurrent unit (BGRU) is applied in *Sadgali, Sael & Benabbou (2021)*. Algorithms such as, GRU, LSTM, BRU, and SMOTE utilised in this model. BGRU obtained a high accuracy of 97.16%.

Long short-term memory (LSTM) is helpful technique to predict fraud because of the history knowledge it contains and the link that exists between prediction outputs and historical input. LSTM architecture enables sequence prediction problems to be learned through long-term reliance. LSTM and BiLSTM applied in eight articles (*Agarwal et al., 2021*; *Alghofaili, Albattah & Rassam, 2020*; *Benchaji, Douzi & El Ouahidi, 2021*; *Cheon et al., 2021*; *Forough & Momtazi, 2021*; *Nguyen et al., 2020*; *Osegi & Jumbo, 2021*; *Sadgali, Sael & Benabbou, 2021*). In *Alghofaili, Albattah & Rassam (2020)*, a new model developed to improve both the present detection techniques and the detection accuracy in light of huge data. Findings demonstrated that LSTM performed perfectly, achieving 99.95% accuracy. *Benchaji, Douzi & El Ouahidi (2021)* recommended a model with the purpose of recording the previous purchasing behaviour of card holders. The results show that LSTM model obtained a high level of performance and accuracy.

DL based hybrid approach of detecting fraudulent transactions applied in *Cheon et al. (2021)*. The new model includes a Bi-LSTM-autoencoder with isolation forest. This model proposed a detection rate of 87% for fraudulent transactions. The suggested model scored the highest mark. This model has the potential to be employed as an effective method for detecting fraud.

Deep belief network (DBN) applied in one article (*Zhang et al., 2021*). The new model utilised DBN and advanced feature engineering base on a Homogeneity-oriented behaviour analysis (HOBA). Results indicate that suggested model is effective and capable to identify fraud. DBN classifier with HOBA achieves a performance that is superior to that of the standard models.

Boltzmann machine (RBM) comprises of visible and hidden layers linked by symmetric weights. The neurones in the visible layer correspond to the X inputs, whilst the responses of the neurones H in hidden layer reflect the eventuality distribution of the inputs. RBM appeared in three articles in the review (*Niu, Wang & Yang, 2019*; *Suthan, 2021*; *Suvarna & Kowshalya, 2020*). In *Niu, Wang & Yang (2019)*, supervised and unsupervised techniques have been applied. XGB and RF as a supervised technique obtain the best performance with AUROC is 0.961. RBM provides the best performance among unsupervised techniques. Results indicate that supervised models outperform the unsupervised models. Because of the problem of inadequate annotation and data imbalance, unsupervised techniques remain promising for credit card fraud detection.

A generative network (GAN) is comprised of two feed forward neural network, a Generate and a Discriminator, competing each other. The G produces new candidates while the D evaluates the quality. Each of the two networks is typically a DNN with multiple layers interconnected. GAN appeared in seven articles (*Ba, 2019*; *Fiore et al., 2019*; *Tingfei, Guangquan & Kuihua, 2020*; *Hwang & Kim, 2020*; *Niu, Wang & Yang, 2019*; *Wu, Cui & Welsch, 2020*; *Veigas, Regulagadda & Kokatnoor, 2021*). In *Ba (2019)*, GANs employed as an oversampling technique. The findings indicate that Wasserstein-GAN is

reliable during training and creates accurate fraudulent transactions comparing with other GANs. In *Fiore et al. (2019)*, GAN employed to enhance the effectiveness of classification. A model for addressing the problem of class imbalance is described. GAN trained to generate minority class instances, then combined with training data to create an augmented training set to enhance performance. The results indicate that a classifier trained on expanded data outperforms its original equivalent.

The input-output mapping between the encoding and decoding phases is discovered by the autoencoder (AE). The input is mapped by the encoder to the hidden layer, and the input is rebuilt by the decoder using the hidden layer as the output layer. AE appeared in 18 articles in this review. AE mentioned in 18 articles within this review. In *Misra et al. (2020)*, autoencoder model for cyber fraud detection is applied. Two-stage model with an autoencoder that coverts the transaction characteristics to a lower-dimensional feature vector at the first step. A classifier is then fed these feature vectors in a subsequent step. Results show that the suggested model outperform other models.

In *Wu, Cui & Welsch (2020)*, dual autoencoders generative adversarial networks (DAEGAN) is employed for the imbalanced classification problem. The new model trains GAN to duplicate fraudulent transaction for autoencoder training. To create two sets of features, two autoencoders encode the samples. The new model outperforms several classification algorithms. Due to extremely skewed class distributions, credit card datasets present classification situations that are unbalanced. To address this difficulty. New model proposes in *Tingfei, Guangquan & Kuihua (2020)* employing oversampling technique based on variational automatic coding (VAE) in combination with DL techniques. Results demonstrate that the VAE model outperforms synthetic minority oversampling strategies and conventional DNN methods. In addition, it performs better than previous oversampling techniques based on GAN models.

### Metaheuristic techniques

In *Makolo & Adeboye (2021)*, a new hybrid model is created by applying Genetic algorithm and multivariate normal distribution to unbalanced dataset. After trained on the same dataset, the prediction accuracy compared to that of DT, ANN, and SVM. The model yielded a remarkable F-score of 93.5%, whereas ANN is 68.5%, DT is 80.0%, and SVM is 84.2%. Enhanced hybrid system for credit card fraud prediction in *Nwogu & Nwachukwu (2019)*. The genetic algorithm with RF model optimisation (GAORF) is employed. Utilising real and genetic algorithms. This model's classification accuracy enhanced through the optimisation of RF models. This can assist in resolving the problem of a shortage of transaction data, as well as the problem of inadequate optimisation and convergence of RF algorithms. The model improved significantly reducing the overall number of misclassifications.

The use of harmony search algorithm (HAS) with NN to increase fraud detection is described in *Daliri (2020)*. The model uses HAS to optimise the parameters of ANN. Proposed NNHS model provides a method based on HAS that successfully predicts the optimal structure for ANN and identifies the algorithm hidden inside the data. The comparisons revealed that the highest accuracy achieved is 86%.

### Instance-based learning

In *Hussein, Abbas & Mahdi (2021)*, fraud detection model utilising various ML algorithm, including NB, DR, rules classifier, lazy classifier (IBK, LWL, and KStar), meta classifier, and function classifier, implemented in this study. Results indicate that lazy classifier (LMT) technique is the most accurate, with an accuracy of 82.086%.

### Percentage of articles that address supervised, unsupervised, or semi-supervised in credit fraud detection?

This section answers RQ2 which attempts to show the proportion of gathered research article that employ supervised, unsupervised, or semi-supervised techniques. We examined credit card fraud detection techniques described in research article. According to Fig. 4, 74% of the chosen article utilised the supervised technique. Consequently, supervised technique is the most commonly employed in the reviewed article. In contrast, 12% utilised unsupervised techniques, and 12% utilised both supervised and unsupervised techniques. A total of 2% of reviewed article utilised semi-supervised learning. Additionally, 1% utilised reinforcement learning. Supervised and unsupervised learning have been implemented in 2019, 2020, and 2021. While semi-supervised learning only implemented three times in 2021. In the same manner, reinforcement learning has only been utilised in 2021. Compared to supervised and unsupervised learning, semi-supervised learning and reinforcement learning were not embraced by a large number of researchers. The ML/DL techniques type of each study article is listed in Table C1 for more information. The proportion of supervised, unsupervised, and semi-supervised is showed in Fig. 4.

### Overall performance estimation of ML/DL model in credit fraud detection

This section addresses RQ3, which concerns the estimate of ML/DL model performance. Accuracy of estimation is the primary performance indicator for ML/DL models. This question focuses on the following features of estimating accuracy; performance metric, accuracy value, and dataset. As the construction of ML/DL models is dependent on the dataset, we examined the data sources of ML/DL models in the reviewed article. In addition, we found a number of datasets utilised in the experiments of associated article. This review articles employs two sets of datasets; real-word data set and synthetic dataset. The dataset utilised most frequently in the reviewed article is a real-word dataset. In addition, 154 research article employed real-world datasets, eight utilised synthetic datasets, and 19 did not specify the dataset source.

Evaluation metrics were used to calculate ML/DL model performance. Confusion matrix provides output matrix that characterises the model's overall effectiveness. ML/DL model's accuracy is compared using confusion matrix sensitivity and specificity, F-score, precision, receiver operating characteristic (ROC), and area under precision recall area (AUPR).

In this review, a number of different performance indicators have been used in addition to accuracy. As shown in Table C1, we found 177 article that clearly presented the performance metrics of the proposed models. Four article did not mention the

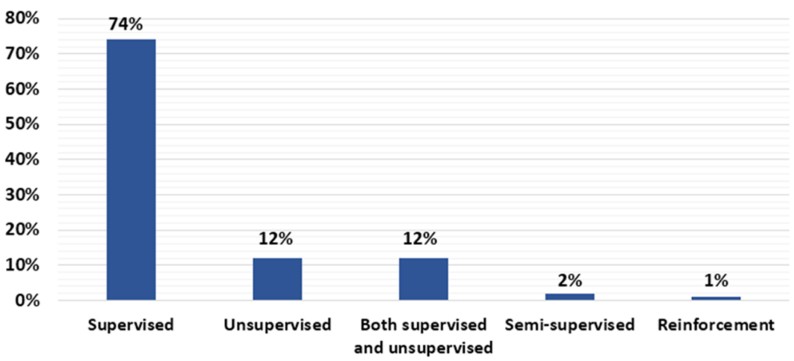

**Figure 4 Percentage of supervised, unsupervised, or semi-supervised.**

performance metrics. We discovered that 177 of reviewed article mentioned the performance indicators of their suggested models. However, four reviewed article did not mention the performance metrics. In this review, accuracy, recall, precision, and F-score were often employed as performance indicators. Accuracy is the proportion of test set records that were properly categorised transaction to fraudulent or non-fraudulent. The ration of true positives to all positives is referred to as precision. The proportion of fraudulent transactions that we correctly detected as fraudulent compared to the total number of fraudulent transactions would be the precision. Recall is percentage of all correctly classified predictions made by an algorithm. In addition, the value of F1 provides a single score that is proportionate to both recall and precision. Full two-dimensional area under the entire ROC curve is measured by AUC. One of the best indicators for analysing the effectiveness of credit card fraud detection is the ROC curve. The classification's quality is measured by MCC. Because it covers true positive, true negatives, false positive, and false negatives, it is a balanced metric. MCC utilised in 13 reviewed article.

In addition, 30 of the 181 studies employed only a single performance metric, with the majority of these article using only accuracy (24) article, MCC (five) article, and execution time (one) article. Using single performance metric is insufficient for determining the quality of ML/DL model. However, article such as 43 and 74 utilised more than five performance indicators to represent the performance of their ML/DL model. In addition, a number of reviewed article give computational performance measurements as well as performance metrics. The length of time the model took to complete the assigned task is called execution time. To ascertain how long the model takes to detect fraud, the execution time is calculated. As a result, we guarantee that the model successfully achieves its goal. Execution time employed in *Alghofaili, Albattah & Rassam (2020)*, *Devi, Thangavel & Anbhazhagan (2019)*, *Singh, Ranjan & Tiwari (2021)*. The loss rate function compares actual and expected training output to speed up learning. Loss rate employed in article (*Alghofaili, Albattah & Rassam, 2020*). Test of the effect of cost sensitive wrapping of Bayes

minimal risk (BMR) applied in article (*Almhaithawi, Jafar & Aljnidi, 2020*) as a cost-saving measure. Balanced accuracy (BCR) combines the matrices of sensitivity and specificity to produce a balanced outcome. BCR presented in article (*Layek, 2020*). In (*Arun & Venkatachalapathy, 2020*) Kappa assesse the predication performance of the classifier model. Few article (*Arya & Sastry, 2020*; *Bandyopadhyay et al., 2021*; *Bandyopadhyay & Dutta, 2020*; *Benchaji, Douzi & El Ouahidi, 2021*; *Rezapour, 2019*) introduced mean square error (MSE) assessment metrics, mean absolute error (MAE), and root mean square error (RMSE). Table C1 shows the proposed ML/DL model along with performance and datasets.

## Trend of research

To answer RQ4, we examine the trend of the reviewed article. In addition, we compare the models created over the three years to determine and evaluate which techniques recently garnered more attention. This also assist, to identify the gaps so that future research will be able to address them in their own work. First, we examined the distribution of the chosen article by the publication year. In year 2019 (47 articles), 2020 (70 articles), and 2021 (64 articles). Significant difference existed between the years 2019 and 2020, the number of published articles for credit card fraud detection increased (23 articles). However, there was no notable difference between 2020 and 2021 (six articles). Fig. 2 demonstrates this comparison.

In response to RQ1, we demonstrated that 110 distinct ML models, 34 distinct DL models, and 39 models that combine ML and DL have been utilised by researchers. RF, LR, and SVM are the most commonly employed ML approaches. ANN, AUE, and LSTM are the most utilised DL approaches. In addition, we observed increased interest in combining ML and DL models.

In our review, we count the various learning-based credit card cyber fraud detection techniques applied in the reviewed article to answer RQ2. From this review we found that the most common technique among the reviewed article is the use of supervised algorithm. Supervised algorithms applied in 74% of the reviewed article. A total of 12% of the reviewed article utilised unsupervised techniques. A total of 12% used supervised and unsupervised techniques. A total of 2% applied semi-supervised technique. A total of 1% used reinforcement technique. For the RQ3, we listed the performance metrics that each research article applied. We discovered that 24 out of 181 reviewed article utilised accuracy as their only key performance metric. We also found a number of datasets that utilised in the reviewed article. Majority of the reviewed article using real-world datasets. A total of 154 research article applied real-world data, eight article used synthetic data, and 19 did not mention the source.

In RQ4, we identified research gaps by investigating unexplored or infrequently studied algorithms. In addition, we found supervised learning as the most prevalent learning

technique and SMOTE as the most prevalent oversampling technique. The majority of researchers focused on supervised techniques such as LR, RF, SVM, and NN.

Combination techniques that employ multiple algorithms are becoming increasingly prevalent in the detection of cyber fraud. Detecting cyber fraud in credit card increasingly involves the use of DL. DL techniques utilised 34 times in the reviewed article, whereas 39 of the reviewed article applied a combination of DL and ML techniques for credit card cyber fraud detection. DL is advantageous for fraud detection since it solves the difficulty of recognising unexpected and sophisticated fraud patterns. Moreover, as the number of fraud cases to be recognised is relatively limited, DL may be effective. DL have garnered the most attention and had the most success in combating cyberthreats recently. Due to its ability to minimise overfitting and discover underlying fraud tendencies. Moreover, the capacity to handle massive datasets.

For supervised learning algorithms to predict future credit card transaction, each observation must have a label. Given that there is no classification for these observations, this could be a problem when trying to identify fraudulent transactions. Additionally, since fraudsters constantly alter their behaviour, it is challenging to develop a supervised learning model for a given transaction. The normal class is often the only one that unsupervised algorithms need labels for, and they can predict future observations based on deviations from the normal data. Future research should give more attention to unsupervised and semi supervised techniques, which can yield new insights. In addition, paying more attention to DL techniques such as CNN, RNN, and LSTM, we recommend that further research may be conducted on ML techniques, especially semi-supervised and unsupervised techniques in order to improve ML model performance. In addition, performing additional research on DL techniques is needed. As a result of the unavailability of a balanced dataset and the shortage of datasets, financial institutions are encouraged to make the essential dataset available, so that research outputs will be more effective and qualitative.

To detect cyber fraud in credit card, supervised, unsupervised, and semi-supervised ML/DL techniques applied in the reviewed article. Figure 4 displays that 74% of the reviewed article utilised supervised techniques. As a result, it is the most common technique used in the reviewed article. In addition, according to the reviewed article, classification and regression techniques been always of interest. On the other hand, 12% of selected articles applied unsupervised techniques, 12% of selected articles applied both supervised and unsupervised techniques, while 2% articles applied semi supervised techniques, and 1% articles applied reinforcement learning. A growing trend in this field is the use of ensemble techniques that capitalise on the benefits of several classification methods. The use of ensemble methods increased in 2020 and 2021 comparing with 2019. The other interesting finding is that DL approaches have attracted considerable interest during 2019 to 2021. The number of research articles that used DL techniques as single technique or combined with other ML techniques in 2019 is 15 articles, in 2020, 30 articles, and in 2021, 28 articles. It appears that the popularity of DL algorithms has increased.

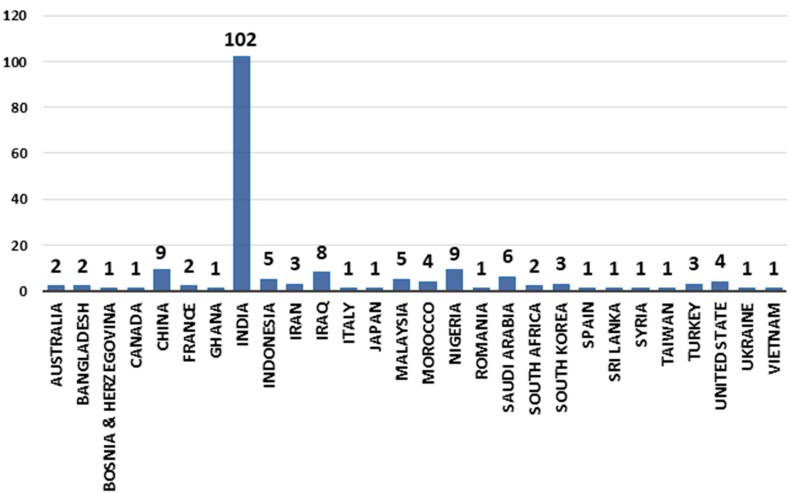

**Figure 5** Articles by country for years (2019, 2020, and 2021).

The countries that published research on utilising ML/DL techniques to detect credit card cyber fraud is growing over time. In 2021, Ghana, Romania, Taiwan, and Vietnam are among the new countries that made an effort in detecting cyber fraud. India is the pioneer when it comes to the publication of ML/DL studies. Figure 5 depicts the number of article published by country and year (2019, 2020, and 2021).

## Gap analysis and the future direction

The most effective way for determining the approaches that are most appropriate for this research problem is to categorise the ML/DL algorithms used in detecting cyber fraud in credit card. Additionally, it is beneficial to determine why particular tactics were chosen. Supervised algorithms have always been of interest, as 74% of the reviewed articles have been used supervised algorithms, with the most commonly used being RF then LR then SVM. Unsupervised learning algorithms also applied in 12% articles with the most commonly used being Isolation forest. However, it is interesting that only 12% of the 181 reviewed studies utilised unsupervised learning techniques. Semi-supervised approach employed in 2% of the reviewed articles. It appears that semi-supervised and unsupervised learning techniques may be researched further. According to reviewed articles (*Choubey & Gautam, 2020*; *More et al., 2021*; *Muaz, Jayabalan & Thiruchelvam, 2020*; *Shirgave et al., 2019*), unsupervised or semi-supervised learning techniques such as one-SVM, isolation forest, and K-means clustering should be utilised more in credit card fraud detection.

In the three years, DL techniques have been examined increasingly frequently. Utilising DL to get greater accuracy and efficient performance. By applying DL techniques, new fraudulent patterns can be recognised and system can respond flexibly to complex data patterns. Thus, for efficient credit card fraud detection, researchers are encouraged to

conduct additional study on DL techniques. Several studies such as (*Benchaji, Douzi & El Ouahidi, 2021*; *Jonnalagadda, Gupta & Sen, 2019*; *Kalid et al., 2020*) suggested further study of DL techniques for detection in credit card. Moreover, as each ML/DL technique has its own limitations, it is necessary to consider combining the ML and DL algorithms for promising detection results. Several article such as (*Agarwal, 2021*; *Dang et al., 2021*; *Gamini et al., 2021*; *Kalid et al., 2020*; *Singh & Jain, 2019*) suggested combinations of DL methods and traditional ML methods to cyber fraud detection from an unbalanced data and enhance the accuracy.

Several reviewed article cited the lack of the dataset as the limitation of their work. According to *Meenu et al. (2020)*, the research outcomes will be more effective and of higher quality if the financial institutions make the crucial data set of various fraudulent actions available. As a result, one of the key problems in many studies is the lack of data. Limitations on the availability of the data could be overcome if there is a vital data set of diverse fraudulent activities across nations. *Maniraj et al. (2019)* noted that when dataset size increase, algorithm precision also increases. It appears that adding additional data will undoubtedly increase the model's ability to detect fraud and decrease the number of false positives. The banks themselves must formally support this. The study (*Seera et al., 2021*) proposed conducting further evaluation of their generated model with real data from diverse regions.

Additionally, the datasets are significantly skewed, which is a problem. Numerous studies attempted to develop a model that could perform properly with data that is highly skewed. Several articles (*Balne, Singh & Yada, 2020*; *Ojugo & Nwankwo, 2021*; *Shekar & Ramakrisha, 2021*; *Voican, 2021*; *Vengatesan et al., 2020*), unbalanced data was applied, and balancing the dataset using sampling techniques such as oversampling or undersampling is left as future work. Several articles (*Ahirwar, Sharma & Bano, 2020*; *Almhaithawi, Jafar & Aljnidi, 2020*; *Manlangit, Azam & Shanmugam, 2019*) applied oversampling techniques.

Undersampling techniques have been applied in several article (*Amusan et al., 2021*; *Ata & Hazim, 2020*; *Muaz, Jayabalan & Thiruchelvam, 2020*; *Rezapour, 2019*; *Zhang, Bhandari & Black, 2020*). In *Amusan et al. (2021)*, a random undersampling technique was used, and the study recommended that other balancing data techniques be explored. One reviewed article (*Ata & Hazim, 2020*) applied an undersampling technique. However, the study recommends adopting the suggested model by using massive dataset instead of using sampling technique. In addition, some articles such as *Trisanto et al. (2021)* and *Singh, Ranjan & Tiwari (2021)* applied undersampling techniques and oversampling techniques.

Oversampling technique such as SMOTE, ADASYN, DBSMOTE, and SMOTEEN have been used. Undersampling techniques such as random undersampling (RUS) has been applied. In light of this, future studies should consider applying alternative oversampling techniques, such as borderline-SMOTE and borderline oversampling with SVM, as well as undersampling techniques. In addition to fraud location, an algorithm to determine the timing of the fraud is required (*Alghofaili, Albattah & Rassam, 2020*; *Chen & Lai, 2021*). In

addition, an algorithm can be developed to predict fraudulent transactions in a real-time and deploying the service on various cloud platforms to make it easily accessible and reliable (*Ingole et al., 2021*).

### Limitation of the review

Our review is restricted to journal article published in 2019, 2020, and 2021 that apply ML/DL techniques. By using our methodology in the early stages, we eliminated several irrelevant article. This assured that the selected article met the requirements for our review. Even though we searched the most prominent digital libraries for the article, there may be more digital libraries having relevant research article that were not included for this study. The snowballing method used to include relevant article that excluded during automatic searching in order to address this limitation. In addition, as it is probable that while looking for the keywords, we would have missed some synonyms. Hence, we also analysed the search terms and keywords for recognised collection of research works. We restricted our search to only English-language articles. This creates a language bias, as there may be article in this field of study written in other languages.

## CONCLUSIONS

This review studied cyber fraud detection in credit card using ML/DL techniques. We examined ML/DL models from the perspectives of ML/DL technique type, ML/DL performance estimation, and the learning-based fraud detection. The study focused on relevant studies that were published in 2019, 2020, and 2021. In order to address the four research questions posed in this study, we reviewed 181 research article. In our review, we have provided a detailed analysis of ML/DL techniques and their function in credit card cyber fraud detection and also offered recommendations for selecting the most suitable techniques for detecting cyber fraud. The study also includes the trends of research, gaps, future direction, and limitations in detecting cyber fraud in credit cards. We believe that this comprehensive review enables researchers and banking industry to develop innovation systems for cyber fraud detection.

On the basis of this analysis, we suggest that more research may be conducted on semi-supervised learning and unsupervised learning techniques. Based on our review, we recommend that DL techniques might be further researched for credit card cyber fraud detection. Researchers are encouraged to conduct further research on integrating the ML/DL algorithms for effective detection outcomes. In addition, researchers are advised to use both oversampling and undersampling techniques because the datasets are extremely skewed. Furthermore, we recommend researchers to mention dataset sources and performance metrics employed to present the outcomes. Banks are also encouraged to make available dataset of different fraudulent activities across nation for further research.

## APPENDIX

### Appendix A

**Table A1 Selected research articles.**

| Article ID | Article title | Type | Year | Reference |
|---|---|---|---|---|
| A1 | Comparative analysis of back-propagation neural network and k-means clustering algorithm in fraud detection in online credit card transaction. | Journal | 2019 | Abdulsalami et al. (2019) |
| A2 | Credit card fraud detection using machine learning classification algorithms over highly imbalanced data. | Journal | 2020 | Adityasundar et al. (2020) |
| A3 | Hybrid CNN-BILSTM-Attention based identification and prevention system for banking transactions. | Journal | 2021 | Agarwal et al. (2021) |
| A4 | Identify theft detection using machine learning. | Journal | 2021 | Agarwal (2021) |
| A5 | Hidden Markov model application for credit card fraud detection systems. | Journal | 2020 | Agbakwuru & Elei (2021) |
| A6 | Enhanced SMOTE & fast random forest techniques for credit card fraud detection. | Journal | 2020 | Ahirwar, Sharma & Bano (2020) |
| A7 | Fraud identification of credit card using ML techniques. | Journal | 2020 | Akula (2020) |
| A8 | Improvement in credit card fraud detection using ensemble classification technique and user data. | Journal | 2021 | Al Rubaie (2021) |
| A9 | Credit card fraud detection *via* integrated account and transaction sub modules. | Journal | 2021 | Al-Faqeh et al. (2021) |
| A10 | Credit card fraud detection using autoencoder model in unbalanced datasets. | Journal | 2019 | Al-Shabi (2019) |
| A11 | Fraud detection in credit card using logistic regression. | Journal | 2020 | Alenzi & Aljehane (2020) |
| A12 | A financial fraud detection model based on LSTM deep learning technique. | Journal | 2020 | Alghofaili, Albattah & Rassam (2020) |
| A13 | Comparative study of machine learning algorithms and correlation between input parameters. | Journal | 2019 | Alias, Ibrahim & Zin (2019) |
| A14 | Example-dependent cost-sensitive credit cards fraud detection using SMOTE and Bayes minimum risk. | Journal | 2020 | Almhaithawi, Jafar & Aljnidi (2020) |
| A15 | Credit card fraud detection on skewed data using machine learning techniques. | Journal | 2021 | Amusan et al. (2021) |
| A16 | Facilitating user authorization from imbalanced data logs of credit card using artificial intelligence. | Journal | 2020 | Arora et al. (2020) |
| A17 | Intelligence feature selection with social spider optimization based artificial neural network model for credit card fraud detection. | Journal | 2020 | Arun & Venkatachalapathy (2020) |
| A18 | Deal-deep ensemble algorithm framework for credit card fraud detection in real-time data stream with Google TensorFlow. | Journal | 2020 | Arya & Sastry (2020) |
| A19 | Credit card fraud detection using artificial neural network. | Journal | 2021 | Asha & Suresh Kumar (2021) |
| A20 | IFDTC4.5: intuitionistic fuzzy logic based decision tree for E-transactional fraud detection. | Journal | 2020 | Askari & Hussain (2020) |
| A21 | Credit card fraud detection using hybrid models. | Journal | 2019 | Aswathy & Samuel (2019) |
| A22 | Comparative analysis of different distribution dataset by using data mining techniques on credit card fraud detection. | Journal | 2020 | Ata & Hazim (2020) |
| A23 | Improving detection of credit card fraudulent transaction using generative adversarial networks. | Journal | 2019 | Ba (2019) |
| A24 | Credit card fraud detection using pipeling and ensemble learning. | Journal | 2020 | Bagga et al. (2020) |
| A25 | Emerging approach for detection of financial fraud using machine learning. | Journal | 2021 | Bandyopadhyay et al. (2021) |
| A26 | Detection of fraud transactions using recurrent neural network during COVID-19: fraud transaction during COVID-19. | Journal | 2020 | Bandyopadhyay & Dutta (2020) |
| A27 | Enhancing the credit card fraud detection through ensemble techniques. | Journal | 2019 | Barahim et al. (2019) |
| A28 | Credit card fraud detection using data mining and statistical methods. | Journal | 2020 | Beigi & Amin Naseri (2020) |
| A29 | Credit card fraud detection model based on LSTM recurrent neural networks. | Journal | 2021 | Benchaji, Douzi & El Ouahidi (2021) |
| A30 | Credit card fraud detection using machine learning algorithms. | Journal | 2020 | Dornadula & Geetha (2019) |
| A31 | Credit card fraud detection using autoencoders. | Journal | 2020 | Balne, Singh & Yada (2020) |
| A32 | Credit card fraud detection using naïve Bayes and robust scaling techniques. | Journal | 2021 | Borse, Patil & Dhotre (2021) |

| Article ID | Article title | Type | Year | Reference |
|---|---|---|---|---|
| A33 | A closer look into the characteristics of fraudulent and transactions. | Journal | 2020 | *Can et al. (2020)* |
| A34 | Evaluation of deep neural networks for reduction of credit card fraud alerts. | Journal | 2020 | *Carrasco & Sicilia-Urbán (2020)* |
| A35 | Deep convolution neural network model for credit-card fraud detection and alert. | Journal | 2021 | *Chen & Lai (2021)* |
| A36 | Graph neural network for fraud detection *via* spatial-temporal attention. | Journal | 2020 | *Cheng et al. (2020)* |
| A37 | Deep learning-based hybrid approach of detecting fraudulent transactions. | Journal | 2021 | *Cheon et al. (2021)* |
| A38 | Combined technique of supervised classifier for the credit card fraud detection. | Journal | 2020 | *Choubey & Gautam (2020)* |
| A39 | Supervised machine learning algorithms for detection credit card fraud. | Journal | 2021 | *Chowdari (2021)* |
| A40 | Using harmony search algorithm in neural networks to improve fraud detection in banking system. | Journal | 2020 | *Daliri (2020)* |
| A41 | Detecting electronic banking fraud on highly imbalanced data using hidden Markov models. | Journal | 2021 | *Danaa, Daabo & Abdul-Barik (2021)* |
| A42 | Machine learning based on resampling approaches and deep reinforcement learning for credit card fraud detection systems. | Journal | 2021 | *Dang et al. (2021)* |
| A43 | Credit card fraud detection system using data mining. | Journal | 2020 | *Das et al. (2020)* |
| A44 | A comparative study on credit card fraud detection. | Journal | 2021 | *Deb, Ghosal & Bose (2021)* |
| A45 | Supervised machine learning algorithms for credit card fraudulent transaction detection. | Journal | 2019 | *DeepaShree et al. (2019)* |
| A46 | Credit card fraud detection analysis using robust space invariant artificial neural networks (RSIANN). | Journal | 2019 | *Deepika & Senthil (2019)* |
| A47 | Credit card fraud detection system. | Journal | 2020 | *Deshmukh et al. (2020)* |
| A48 | Artificial intelligence based credit card fraud identification using fusion method. | Journal | 2019 | *Devi, Thangavel & Anbhazhagan (2019)* |
| A49 | Credit card fraud detection using random forest. | Journal | 2019 | *Meenakshi et al. (2019)* |
| A50 | Performance evaluation of credit card fraud transaction using boosting algorithms. | Journal | 2019 | *Divakar & Chitharanjan (2019)* |
| A51 | Fraud detection in credit card transaction using anomaly detection. | Journal | 2021 | *Dwivedi (2021)* |
| A52 | Semi-supervised classification on credit card fraud detection using autoencoders. | Journal | 2021 | *Dzakiyullah, Pramuntadi & Fauziyyah (2021)* |
| A53 | Artificial neural network technique for improving predication of credit card default: a stacked sparse autoencoder approach. | Journal | 2021 | *Ebiaredoh-Mienye, Esenogho & Swart (2021)* |
| A54 | Credit card fraud detection based on machine learning. | Journal | 2019 | *Fang, Zhang & Huang (2019)* |
| A55 | Comparison of different ensemble methods in credit card default prediction. | Journal | 2021 | *Faraj, Mahmud & Rashid (2021)* |
| A56 | A novel method for detection of fraudulent bank transactions using multi-layer neural networks with adaptive learning rate. | Journal | 2020 | *Faridpour & Moradi (2020)* |
| A57 | Using generative adversarial networks for improving classification effectives in credit card fraud detection. | Journal | 2019 | *Fiore et al. (2019)* |
| A58 | Ensemble of deep sequential models for credit card fraud detection. | Journal | 2021 | *Forough & Momtazi (2021)* |
| A59 | Detection of credit card fraudulent transaction using boosting algorithms. | Journal | 2021 | *Gamini et al. (2021)* |
| A60 | Predication credit card transaction fraud using machine learning algorithms. | Journal | 2019 | *Gao et al. (2019)* |
| A61 | Financial fraud detection using naïve Bayes algorithm in highly imbalance data set. | Journal | 2021 | *Gupta, Lohani & Manchanda (2021)* |
| A62 | Anomaly detection in credit card transactions using machine learning. | Journal | 2020 | *Meenu et al. (2020)* |
| A63 | Uncertainty-aware credit card fraud detection using deep learning. | Journal | 2021 | *Habibpour et al. (2021)* |
| A64 | Credit card fraud detection using ensemble classifier. | Journal | 2019 | *Hameed & RamKumar (2019)* |
| A65 | An implementation of decision tree algorithm augmented with regression analysis for fraud detection in credit card. | Journal | 2020 | *Hammed & Soyemi (2020)* |

(Continued)

| Article ID | Article title | Type | Year | Reference |
|---|---|---|---|---|
| A66 | Credit card fraud detection technique using hybrid approach: an amalgamation of self-organizing maps and neural networks. | Journal | 2020 | *Harwani et al. (2020)* |
| A67 | Machine learning methods for discovering credit card fraud. | Journal | 2020 | *Hema & Muttipati (2020)* |
| A68 | Improved deep forest more for detection of fraudulent online transaction. | Journal | 2020 | *Huang, Wang & Zhang (2020)* |
| A69 | Using variational auto encoding in credit card fraud detection. | Journal | 2020 | *Tingfei, Guangquan & Kuihua (2020)* |
| A70 | Credit card fraud detection using naïve Bayesian and c4.5 decision tree classifiers. | Journal | 2020 | *Husejinovic (2020)* |
| A71 | Credit card fraud detection using fuzzy rough nearest neighbor and sequential minimal optimization with logistic regression. | Journal | 2021 | *Hussein et al. (2021)* |
| A72 | Fraud classification and detection model using different machine learning algorithm. | Journal | 2021 | *Hussein, Abbas & Mahdi (2021)* |
| A73 | An efficient domain-adaptation method using different machine learning GAN for fraud detection. | Journal | 2020 | *Hwang & Kim (2020)* |
| A74 | Service-based credit card fraud detection using oracle SOA suite. | Journal | 2021 | *Ingole et al. (2021)* |
| A75 | Comparison and analysis of logistic regression, naïve Bayes and KNN machine learning algorithms for credit card fraud detection. | Journal | 2021 | *Itoo & Singh (2021)* |
| A76 | Credit card fraud detection using isolation forest and local factor. | Journal | 2021 | *Jaiswal, Brindha & Lakhotia (2021)* |
| A77 | Credit card fraud detection using random forest algorithm. | Journal | 2019 | *Jonnalagadda, Gupta & Sen (2019)* |
| A78 | A multiple classifiers system for anomaly detection in credit card data with unbalanced and overlapped classes. | Journal | 2020 | *Kalid et al. (2020)* |
| A79 | Supervised machine learning algorithms for credit card fraudulent transaction detection. | Journal | 2019 | *Karthik et al. (2019)* |
| A80 | Credit card fraud detection using machine learning. | Journal | 2019 | *Karthikeyan et al. (2019)* |
| A81 | Champion-challenger analysis for credit card fraud detection: hybrid ensemble and deep learning. | Journal | 2019 | *Kim et al. (2019)* |
| A82 | A novel framework for credit card fraud detection. | Journal | 2021 | *Kochhara & Chhabrab (2021)* |
| A83 | Automatic machine learning algorithms for fraud detection in digital payment systems. | Journal | 2020 | *Kolodiziev et al. (2020)* |
| A84 | A new hybrid method for credit card fraud detection on financial data. | Journal | 2019 | *Krishna, Nagini & Tatayyanaidu (2019)* |
| A85 | A study of fraud detection approaches in credit card transactions. | Journal | 2020 | *Rao et al. (2020)* |
| A86 | Credit card fraud detection using Bayesian belief network. | Journal | 2020 | *Kumar, Mubarak & Dhanush (2020)* |
| A87 | An efficient approach for credit card fraud detection. | Journal | 2020 | *Kumar, Student & Budihul (2020)* |
| A88 | Comparative analysis for fraud detection using logistic regression, random forest and support vector machine. | Journal | 2020 | *Kumar, Saini & Payal (2020)* |
| A89 | Fraud detection and prevention in banking financial transaction with machine learning using R. | Journal | 2020 | *Layek (2020)* |
| A90 | Comparative study on credit card fraud detection based on different support vector machines. | Journal | 2021 | *Li et al. (2021)* |
| A91 | Credit card fraud detection with autoencoder and probabilistic random forest. | Journal | 2021 | *Lin & Jiang (2021)* |
| A92 | Towards automated feature engineering for credit card fraud detection using multi-perspective HMMs. | Journal | 2020 | *Lucas et al. (2020)* |
| A93 | An experimental study with imbalanced classification approaches for credit card fraud detection. | Journal | 2019 | *Makki et al. (2019)* |
| A94 | Credit card fraud detection system using machine learning. | Journal | 2021 | *Makolo & Adeboye (2021)* |
| A95 | Analysis of credit card fraud detection using machine learning models on balanced and imbalanced datasets. | Journal | 2021 | *Mallidi & Zagabathuni (2021)* |
| A96 | Credit card fraud detection using machine learning and data science. | Journal | 2019 | *Maniraj et al. (2019)* |

| Article ID | Article title | Type | Year | Reference |
|---|---|---|---|---|
| A97 | Novel machine learning approach for analysis anonymous credit card fraud patterns. | Journal | 2019 | *Manlangit, Azam & Shanmugam (2019)* |
| A98 | Credit card fraud detection using machine learning. | Journal | 2021 | *Marabad (2021)* |
| A99 | Detection fraudulent credit card transactions using outlier detection. | Journal | 2019 | *Marella et al. (2019)* |
| A100 | Credit card fraud detection in payment using machine learning classifiers. | Journal | 2020 | *Mijwil & Salem (2020)* |
| A101 | An autoencoder based model for detecting fraudulent credit card transaction. | Journal | 2020 | *Misra et al. (2020)* |
| A102 | A comparative study on classification algorithms for credit card fraud detection. | Journal | 2020 | *Mohari et al. (2020)* |
| A103 | Credit card fraud detection using random forest algorithm. | Journal | 2019 | *Monika et al. (2019)* |
| A104 | Credit card fraud detection using supervised learning approach. | Journal | 2021 | *More et al. (2021)* |
| A105 | A SOMTE based oversampling data-point approach to solving the credit card data imbalance problem in financial fraud detection. | Journal | 2021 | *Mqadi, Naicker & Adeliyi (2021)* |
| A106 | Using machine learning to detect credit card fraudulent transactions. | Journal | 2021 | *Vijay Rahul et al. (2021)* |
| A107 | Credit card fraud detection using autoencoder neural network. | Journal | 2019 | *Zou, Zhang & Jiang (2019)* |
| A108 | Credit card fraud detection using ANN. | Journal | 2019 | *Oumar & Augustin (2019)* |
| A109 | An improved hybrid system for the prediction of debit and credit card fraud. | Journal | 2019 | *Nwogu & Nwachukwu (2019)* |
| A110 | Deep learning methods for credit card fraud detection. | Journal | 2020 | *Nguyen et al. (2020)* |
| A111 | A comparison of data sampling techniques for credit card fraud detection. | Journal | 2020 | *Muaz, Jayabalan & Thiruchelvam (2020)* |
| A112 | Credit card fraud detection using machine learning algorithms. | Journal | 2020 | *Parashar & Bhati (2020)* |
| A113 | A machine learning approach for detecting credit card fraudulent transaction. | Journal | 2021 | *Nimashini, Rathnayake & Wickramaarachchi (2021)* |
| A114 | Credit card fraud detection using AdaBoost. | Journal | 2020 | *Nithin, Ravula & Sulthana (2020)* |
| A115 | A comparison study of credit card fraud detection: supervise *vs* unsupervised. | Journal | 2019 | *Niu, Wang & Yang (2019)* |
| A116 | Credit card fraud detection using random forest algorithm. | Journal | 2019 | *Niveditha, Abarna & Akshaya (2019)* |
| A117 | A comparative study of machine learning classifiers for credit card fraud detection. | Journal | 2020 | *Nur-E-Arefin (2020)* |
| A118 | Spectral-cluster solution for credit-card fraud detection using a genetic algorithm trained modular deep learning neural network. | Journal | 2021 | *Ojugo & Nwankwo (2021)* |
| A119 | Comparative analysis of credit card fraud detection in simulated annealing trained artificial neural network and hierarchical temporal memory. | Journal | 2021 | *Osegi & Jumbo (2021)* |
| A120 | Credit card fraud detection using isolation forest. | Journal | 2021 | *Singh et al. (2021)* |
| A121 | Credit card fraud detection using machine learning algorithms. | Journal | 2020 | *Varun Kumar et al. (2020)* |
| A122 | Credit card fraud detection framework a machine learning perspective. | Journal | 2020 | *Parmar, Patel & Savsani (2020)* |
| A123 | The improving prediction of credit card fraud detection on PSO optimized SVM. | Journal | 2019 | *Pavithra & Thangadurai (2019)* |
| A124 | Credit card fraud detection using boosted stacking. | Journal | 2019 | *Prabhakara et al. (2019)* |
| A125 | Credit card fraud detection technique by applying graph database model. | Journal | 2021 | *Prusti, Das & Rath (2021)* |
| A126 | Online fraud detection using deep learning techniques. | Journal | 2021 | *Suthan (2021)* |
| A127 | A hybrid method for credit card fraud detection using machine learning algorithm. | Journal | 2021 | *Pratap & Vijayaraghavulu (2021)* |
| A128 | Anomaly detection using unsupervised methods: credit card fraud case study. | Journal | 2019 | *Rezapour (2019)* |
| A129 | Discovering of credit card scheme with enhance and common by vote. | Journal | 2021 | *Roy & Rasheeduddin (2021)* |
| A130 | Enhanced credit card fraud detection based on SVM-recursive feature elimination and hyper-parameters optimization. | Journal | 2020 | *Rtayli & Enneya (2020)* |
| A131 | Bidirectional gated recurrent unit for improving classification on credit card fraud detection. | Journal | 2021 | *Sadgali, Sael & Benabbou (2021)* |

(Continued)

| Article ID | Article title | Type | Year | Reference |
|---|---|---|---|---|
| | **Table A1** *(continued)* | | | |
| A132 | Credit card fraud detection system using smote technique and whale optimization algorithm. | Journal | 2019 | *Sahayasakila, Aishwaryasikhakolli & Yasaswi (2019)* |
| A133 | Fraud detection in online transaction. | Journal | 2020 | *Sahoo et al. (2020)* |
| A134 | Credit card fraud detection using machine learning. | Journal | 2021 | *Saiju et al. (2021)* |
| A135 | Machine learning approach on apache spark for credit card fraud detection. | Journal | 2020 | *Santosh & Ramesh (2020)* |
| A136 | Credit card fraud detection using weighted support vector machine. | Journal | 2020 | *Zhang, Bhandari & Black (2020)* |
| A137 | Machine learning methods for analysis fraud credit card transaction. | Journal | 2019 | *Saragih et al. (2019)* |
| A138 | A review on credit card fraud detection using machine learning. | Journal | 2019 | *Shirgave et al. (2019)* |
| A139 | Financial fraud detection using bio-inspired key optimization and machine learning technique. | Journal | 2019 | *Singh & Jain (2019)* |
| A140 | Semisupervised algorithms based credit card fraud detection using majority voting. | Journal | 2021 | *Shekar & Ramakrisha (2021)* |
| A141 | Artificial intelligence framework for credit card fraud detection using supervised random forest. | Journal | 2021 | *Sarvani & Markandeyulu (2021)* |
| A142 | An intelligent payment card fraud detection system. | Journal | 2021 | *Seera et al. (2021)* |
| A143 | HOBA: a novel feature engineering methodology for credit card fraud detection with a deep learning architecture. | Journal | 2021 | *Zhang et al. (2021)* |
| A144 | Dual autoencoders generative adversarial network for imbalanced classification problem. | Journal | 2020 | *Wu, Cui & Welsch (2020)* |
| A145 | Performance analysis of isolation forest algorithm in fraud detection of credit card transactions. | Journal | 2020 | *Waspada et al. (2020)* |
| A146 | Credit card fraud detection from imbalanced dataset using machine learning algorithm. | Journal | 2020 | *Warghade, Desai & Patil (2020)* |
| A147 | Credit card fraud forecasting model based on clustering analysis and integrated support vector machine. | Journal | 2019 | *Wang & Han (2019)* |
| A148 | Credit card anomaly detection using improved deep autoencoder algorithm. | Journal | 2020 | *Waleed, Mawlood & Jabber Abdulhussien (2020)* |
| A149 | Credit card fraud detection using deep learning techniques. | Journal | 2021 | *Voican (2021)* |
| A150 | Detecting credit card frauds using different machine learning algorithms. | Journal | 2021 | *Visalakshi, Madhuvani & Sunilraja (2021)* |
| A151 | Isolation forest and local outlier factor for credit card fraud detection system. | Journal | 2020 | *Vijayakumar et al. (2020)* |
| A152 | Analysis of machine learning credit card fraud detection models. | Journal | 2021 | *Uloko et al. (2021)* |
| A153 | Time varying inertia weight dragonfly algorithm with weighted feature-based support vector machine for credit card fraud detection. | Journal | 2021 | *Arun & Venkatachalapathy (2021)* |
| A154 | Predicting credit card fraud on a imbalanced data. | Journal | 2019 | *Soh & Yusuf (2019)* |
| A155 | Master card fraud detection using arbitrary forest. | Journal | 2019 | *Sireesha et al. (2020)* |
| A156 | Credit card fraud detection using data analytic techniques. | Journal | 2020 | *Vengatesan et al. (2020)* |
| A157 | Optimized stacking ensemble (OSE) for credit card fraud detection using synthetic minority oversampling model. | Journal | 2021 | *Veigas, Regulagadda & Kokatnoor (2021)* |
| A158 | Aggrandized random forest to detect the credit card frauds. | Journal | 2019 | *Vadakara & Kumar (2019)* |
| A159 | An efficient credit card fraud detection model based on machine learning methods. | Journal | 2020 | *Trivedi et al. (2020)* |
| A160 | Modified focal loss in imbalanced XGBoost for credit card fraud detection. | Journal | 2021 | *Trisanto et al. (2021)* |
| A161 | Credit card fraud detection using hidden Markov model. | Journal | 2019 | *Singh et al. (2019)* |
| A162 | Credit card fraud detection using isolation forest. | Journal | 2020 | *Palekar et al. (2020)* |
| A163 | Comparing different models for credit card fraud detection. | Journal | 2020 | *Keskar (2020)* |

| Article ID | Article title | Type | Year | Reference |
|---|---|---|---|---|
| A164 | Credit card fraud detection under extreme imbalanced data: a comparative study of data-level algorithms. | Journal | 2021 | *Singh, Ranjan & Tiwari (2021)* |
| A165 | Credit card fraud detection: a comparison using random forest, SVM and ANN. | Journal | 2019 | *Simi (2019)* |
| A166 | Credit card fraud detection using machine learning methodology. | Journal | 2019 | *Shukur & Kurnaz (2019)* |
| A167 | Credit card fraud detection using machine learning. | Journal | 2021 | *Sellam et al. (2021)* |
| A168 | An intelligent approach to credit card fraud detection using an optimized light gradient boosting machine. | Journal | 2020 | *Taha & Malebary (2020)* |
| A169 | Real time credit card fraud detection. | Journal | 2021 | *Tadvi et al. (2021)* |
| A170 | Credit card fraud detection using federated learning techniques. | Journal | 2020 | *Suvarna & Kowshalya (2020)* |
| A171 | A supervised learning algorithm for credit card fraud detection. | Journal | 2021 | *Surannagari et al. (2020)* |
| A172 | A comparative study of credit card fraud detection using machine learning for United Kingdom dataset. | Journal | 2019 | *Sujatha (2019)* |
| A173 | Outlier detection credit card transactions using local outlier factor algorithm (LOF). | Journal | 2019 | *Sugidamayatno & Lelono (2019)* |
| A174 | Credit card fraud detection using machine learning approach. | Journal | 2021 | *Soni (2021)* |
| A175 | Real-time deep learning based credit card fraud detection. | Journal | 2020 | *Sobana Devi & Ravi (2020)* |
| A176 | A perceptron based neural network data analysis architecture for the detection of fraud in credit card transactions in financial legacy system. | Journal | 2021 | *Smith & Valverde (2021)* |
| A177 | Credit card fraud detection techniques. | Journal | 2020 | *Shirodkar et al. (2020)* |
| A178 | Adaptive model for credit card fraud detection. | Journal | 2020 | *Sadgali, Sael & Benabbou (2020)* |
| A179 | Credit card fraud detection by modelling behaviour pattern using hybrid ensemble model. | Journal | 2021 | *Karthik, Mishra & Reddy (2022)* |
| A180 | Credit card fraud detection using PSO optimized neural network. | Journal | 2020 | *Dashora, Sharma & Bhargava (2020)* |
| A181 | Detection and prediction of credit card fraud transactions using machine learning. | Journal | 2019 | *Leena & Ajeet (2019)* |

# Appendix B

**Table B1  Usage frequency of ML and DL techniques in credit card fraud.**

| Learning type | Technique | Usage frequency | Reference |
|---|---|---|---|
| Supervised | Logic regression (LR) | 52 | *Adityasundar et al. (2020)*, *Al-Shabi (2019)*, *Alenzi & Aljehane (2020)*, *Almhaithawi, Jafar & Aljnidi (2020)*, *Amusan et al. (2021)*, *Arora et al. (2020)*, *Arya & Sastry (2020)*, *Bagga et al. (2020)*, *Bandyopadhyay et al. (2021)*, *Dornadula & Geetha (2019)*, *Chen & Lai (2021)*, *Choubey & Gautam (2020)*, *Chowdari (2021)*, *Divakar & Chitharanjan (2019)*, *Dwivedi (2021)*, *Faridpour & Moradi (2020)*, *Gao et al. (2019)*, *Gupta, Lohani & Manchanda (2021)*, *Hameed & RamKumar (2019)*, *Hema & Muttipati (2020)*, *Hussein et al. (2021)*, *Itoo & Singh (2021)*, *Karthik et al. (2019)*, *Kim et al. (2019)*, *Kochhara & Chhabrab (2021)*, *Kumar, Student & Budihul (2020)*, *Kumar, Saini & Payal (2020)*, *Layek (2020)*, *Makki et al. (2019)*, *Mallidi & Zagabathuni (2021)*, *Marabad (2021)*, *Misra et al. (2020)*, *Mohari et al. (2020)*, *Mqadi, Naicker & Adeliyi (2021)*, *Oumar & Augustin (2019)*, *Parashar & Bhati (2020)*, *Nimashini, Rathnayake & Wickramaarachchi (2021)*, *Niu, Wang & Yang (2019)*, *Singh et al. (2021)*, *Parmar, Patel & Savsani (2020)*, *Prabhakara et al. (2019)*, *Soh & Yusuf (2019)*, *Vengatesan et al. (2020)*, *Trivedi et al. (2020)*, *Trisanto et al. (2021)*, *Keskar (2020)*, *Singh, Ranjan & Tiwari (2021)*, *Sujatha (2019)*, *Soni (2021)*, *Shirodkar et al. (2020)*, *Karthik, Mishra & Reddy (2022)*, *Leena & Ajeet (2019)*. |

(Continued)

| Learning type | Technique | Usage frequency | Reference |
|---|---|---|---|
| | Naive Bayes (NB) | 42 | *Akula (2020), Aswathy & Samuel (2019), Ata & Hazim (2020), Bagga et al. (2020), Bandyopadhyay et al. (2021), Borse, Patil & Dhotre (2021), Can et al. (2020), Choubey & Gautam (2020), DeepaShree et al. (2019), Deshmukh et al. (2020), Devi, Thangavel & Anbhazhagan (2019), Divakar & Chitharanjan (2019), Gupta, Lohani & Manchanda (2021), Hameed & RamKumar (2019), Husejinovic (2020), Hussein et al. (2021), Hussein, Abbas & Mahdi (2021), Itoo & Singh (2021), Kalid et al. (2020), Karthik et al. (2019), Karthikeyan et al. (2019), Kim et al. (2019), Rao et al. (2020), Kumar, Mubarak & Dhanush (2020), Kumar, Student & Budihul (2020), Makki et al. (2019), Mallidi & Zagabathuni (2021), Mijwil & Salem (2020), Mohari et al. (2020), Nithin, Ravula & Sulthana (2020), Varun Kumar et al. (2020), Roy & Rasheeduddin (2021), Sahoo et al. (2020), Shekar & Ramakrisha (2021), Seera et al. (2021), Visalakshi, Madhuvani & Sunilraja (2021), Trivedi et al. (2020), Trisanto et al. (2021), Singh, Ranjan & Tiwari (2021), Taha & Malebary (2020), Sujatha (2019), Soni (2021)* |
| | Decision tree (DT) | 49 | *Akula (2020), Alias, Ibrahim & Zin (2019), Amusan et al. (2021), Arora et al. (2020), Askari & Hussain (2020), Aswathy & Samuel (2019), Bandyopadhyay et al. (2021), Barahim et al. (2019), Beigi & Amin Naseri (2020), Dornadula & Geetha (2019), Can et al. (2020), Choubey & Gautam (2020), Chowdari (2021), Deshmukh et al. (2020), Devi, Thangavel & Anbhazhagan (2019), Divakar & Chitharanjan (2019), Dwivedi (2021), Hameed & RamKumar (2019), Hammed & Soyemi (2020), Husejinovic (2020), Hussein, Abbas & Mahdi (2021), Jonnalagadda, Gupta & Sen (2019), Kim et al. (2019), Kolodziev et al. (2020), Makolo & Adeboye (2021), Mallidi & Zagabathuni (2021), Marabad (2021), Marella et al. (2019), Mijwil & Salem (2020), Mohari et al. (2020), Mqadi, Naicker & Adeliyi (2021), Parashar & Bhati (2020), Nimashini, Rathnayake & Wickramaarachchi (2021), Niu, Wang & Yang (2019), Singh et al. (2021), Varun Kumar et al. (2020), Parmar, Patel & Savsani (2020), Prabhakara et al. (2019), Prusti, Das & Rath (2021), Roy & Rasheeduddin (2021), Santosh & Ramesh (2020), Seera et al. (2021), Visalakshi, Madhuvani & Sunilraja (2021), Uloko et al. (2021), Soh & Yusuf (2019), Singh, Ranjan & Tiwari (2021), Taha & Malebary (2020), Sujatha (2019), Soni (2021).* |
| | Random forest (RF) | 74 | *Ahirwar, Sharma & Bano (2020), Akula (2020), Al Rubaie (2021), Alias, Ibrahim & Zin (2019), Almhaithawi, Jafar & Aljnidi (2020), Amusan et al. (2021), Arora et al. (2020), Aswathy & Samuel (2019), Ata & Hazim (2020), Bagga et al. (2020), Bandyopadhyay et al. (2021), Dornadula & Geetha (2019), Can et al. (2020), Chen & Lai (2021), Choubey & Gautam (2020), Chowdari (2021), DeepaShree et al. (2019), Meenakshi et al. (2019), Divakar & Chitharanjan (2019), Dwivedi (2021), Fang, Zhang & Huang (2019), Gao et al. (2019), Gupta, Lohani & Manchanda (2021), Hameed & RamKumar (2019), Hema & Muttipati (2020), Hussein et al. (2021), Ingole et al. (2021), Jonnalagadda, Gupta & Sen (2019), Karthik et al. (2019), Rao et al. (2020), Kumar, Saini & Payal (2020), Layek (2020), Lin & Jiang (2021), Mallidi & Zagabathuni (2021), Marabad (2021), Marella et al. (2019), Mohari et al. (2020), Monika et al. (2019), More et al. (2021), Mqadi, Naicker & Adeliyi (2021), Vijay Rahul et al. (2021), Nwogu & Nwachukwu (2019), Muaz, Jayabalan & Thiruchelvam (2020), Parashar & Bhati (2020), Nimashini, Rathnayake & Wickramaarachchi (2021), Niu, Wang & Yang (2019), Niveditha, Abarna & Akshaya (2019), Singh et al. (2021), Varun Kumar et al. (2020), Parmar, Patel & Savsani (2020), Prabhakara et al. (2019), Prusti, Das & Rath (2021), Sahoo et al. (2020), Saiju et al. (2021), Shirgave et al. (2019), Sarvani & Markandeyulu (2021), Seera et al. (2021), Zhang et al. (2021), Visalakshi, Madhuvani & Sunilraja (2021), Uloko et al. (2021), Soh & Yusuf (2019), Sireesha et al. (2020), Vadakara & Kumar (2019), Trivedi et al. (2020), Keskar (2020), Singh, Ranjan & Tiwari (2021), Simi (2019), Sellam et al. (2021), Tadvi et al. (2021), Surannagari et al. (2020), Sujatha (2019), Soni (2021), Karthik, Mishra & Reddy (2022), Leena & Ajeet (2019).* |
| | K-near neighbor (KNN) | 39 | *Akula (2020), Al-Faqeh et al. (2021), Alenzi & Aljehane (2020), Alias, Ibrahim & Zin (2019), Amusan et al. (2021), Arora et al. (2020), Asha & Suresh Kumar (2021), Ata & Hazim (2020), Bagga et al. (2020), Choubey & Gautam (2020), Chowdari (2021), DeepaShree et al. (2019), Itoo & Singh (2021), Kalid et al. (2020), Karthik et al. (2019), Krishna, Nagini & Tatayyanaidu (2019), Rao et al. (2020), Kumar, Student & Budihul (2020), Makki et al. (2019), Mallidi & Zagabathuni (2021), Manlangit, Azam & Shanmugam (2019), Marabad (2021), Misra et al. (2020), Mohari et al. (2020), Niu, Wang & Yang (2019), Singh et al. (2021), Parmar, Patel & Savsani (2020), Prusti, Das & Rath (2021), Singh & Jain (2019), Soh & Yusuf (2019), Vengatesan et al. (2020), Veigas, Regulagadda & Kokatnoor (2021), Trisanto et al. (2021), Keskar (2020), Singh, Ranjan & Tiwari (2021), Shukur & Kurnaz (2019), Taha & Malebary (2020), Sujatha (2019), Soni (2021)* |
| | Support vector machine (SVM) | 56 | *Akula (2020), Al Rubaie (2021), Al-Faqeh et al. (2021), Alias, Ibrahim & Zin (2019), Arora et al. (2020), Arya & Sastry (2020), Asha & Suresh Kumar (2021), Ata & Hazim (2020), Barahim et al. (2019), Beigi & Amin Naseri (2020), Dornadula & Geetha (2019), Chen & Lai (2021), Deshmukh et al. (2020), Devi, Thangavel & Anbhazhagan (2019), Faridpour & Moradi (2020), Gao et al. (2019), Gupta, Lohani & Manchanda (2021), Hwang & Kim (2020), Ingole et al. (2021), Kalid et al. (2020), Karthik et al. (2019), Karthikeyan et al. (2019), Rao et al. (2020), Kumar, Saini & Payal (2020), Layek (2020), Li et al. (2021), Makki et al. (2019), Makolo & Adeboye (2021), Mqadi, Naicker & Adeliyi (2021), Nithin, Ravula & Sulthana (2020), Niu, Wang & Yang (2019), Parmar, Patel & Savsani (2020), Pavithra & Thangadurai (2019), Prusti, Das & Rath (2021), Pratap & Vijayaraghavulu (2021), Rezapour (2019), Roy & Rasheeduddin (2021), Rtayli & Enneya (2020), Sahoo et al. (2020), Saiju et al. (2021), Zhang, Bhandari & Black (2020), Singh & Jain (2019), Shekar & Ramakrisha (2021), Seera et al. (2021), Zhang et al. (2021), Warghade, Desai & Patil (2020), Wang & Han (2019), Arun & Venkatachalapathy (2021), Veigas, Regulagadda & Kokatnoor (2021), Trivedi et al. (2020), Trisanto et al. (2021), Singh, Ranjan & Tiwari (2021), Simi (2019), Taha & Malebary (2020), Soni (2021), Sadgali, Sael & Benabbou (2020).* |
| | Bayesian belief networks | 2 | *Kumar, Mubarak & Dhanush (2020), Makki et al. (2019)* |
| | Genetic algorithm (GA) | 5 | *Li et al. (2021), Mohari et al. (2020), Nwogu & Nwachukwu (2019), Ojugo & Nwankwo (2021), Shirodkar et al. (2020).* |
| | Artificial immune systems (AIS) | 1 | *Makki et al. (2019)* |
| | Fuzzy logic | 1 | *Askari & Hussain (2020)* |
| | Logistic model tree (LMT) | 1 | *Hussein, Abbas & Mahdi (2021)* |

| Learning type | Technique | Usage frequency | Reference |
|---|---|---|---|
| Unsupervised | Hidden Markov model (HMM) | 7 | *Agbakwuru & Elei (2021)*, *Danaa, Daabo & Abdul-Barik (2021)*, *Das et al. (2020)*, *Lucas et al. (2020)*, *Mohari et al. (2020)*, *Uloko et al. (2021)*, *Singh et al. (2019)*. |
| | K-means | 7 | *Abdulsalami et al. (2019)*, *Al Rubaie (2021)*, *Beigi & Amin Naseri (2020)*, *Deb, Ghosal & Bose (2021)*, *Santosh & Ramesh (2020)*, *Wang & Han (2019)*, *Palekar et al. (2020)*. |
| | Isolation forest | 19 | *Dornadula & Geetha (2019)*, *Cheon et al. (2021)*, *Dwivedi (2021)*, *Meenu et al. (2020)*, *Ingole et al. (2021)*, *Jaiswal, Brindha & Lakhotia (2021)*, *Layek (2020)*, *Maniraj et al. (2019)*, *Mohari et al. (2020)*, *Parashar & Bhati (2020)*, *Singh et al. (2021)*, *Prusti, Das & Rath (2021)*, *Saiju et al. (2021)*, *Saragih et al. (2019)*, *Waspada et al. (2020)*, *Warghade, Desai & Patil (2020)*, *Vijayakumar et al. (2020)*, *Palekar et al. (2020)*, *Shukur & Kurnaz (2019)* |
| | Self-organizing map (SOM) | 2 | *Deb, Ghosal & Bose (2021)*, *Harwani et al. (2020)* |
| | Principle component analysis (PCA) | 3 | *Deb, Ghosal & Bose (2021)*, *Layek (2020)*, *Manlangit, Azam & Shanmugam (2019)*. |
| | Density based spatial clustering of applications with noise (DBSCAN) | 1 | *Danaa, Daabo & Abdul-Barik (2021)*, *Mallidi & Zagabathuni (2021)* |
| | Local outlier factor (LOF) | 13 | *Dornadula & Geetha (2019)*, *Dwivedi (2021)*, *Jaiswal, Brindha & Lakhotia (2021)*, *Maniraj et al. (2019)*, *Mohari et al. (2020)*, *Parashar & Bhati (2020)*, *Prusti, Das & Rath (2021)*, *Saiju et al. (2021)*, *Warghade, Desai & Patil (2020)*, *Vijayakumar et al. (2020)*, *Palekar et al. (2020)*, *Shukur & Kurnaz (2019)*, *Sugidamayatno & Lelono (2019)*. |
| | One-class SVM | 3 | *Parashar & Bhati (2020)*, *Niu, Wang & Yang (2019)*, *Rezapour (2019)* |
| Semi-supervised | Semi-supervised learning | 3 | *Dzakiyullah, Pramuntadi & Fauziyyah (2021)*, *Pratap & Vijayaraghavulu (2021)*, *Shekar & Ramakrisha (2021)* |
| Reinforcement | Reinforcement | 1 | *Dang et al. (2021)* |
| Ensemble learning | ADA Boost | 20 | *Akula (2020)*, *Alias, Ibrahim & Zin (2019)*, *Arora et al. (2020)*, *Bagga et al. (2020)*, *Barahim et al. (2019)*, *Beigi & Amin Naseri (2020)*, *Divakar & Chitharanjan (2019)*, *Karthikeyan et al. (2019)*, *Kochhara & Chhabrab (2021)*, *Krishna, Nagini & Tatayyanaidu (2019)*, *Mohari et al. (2020)*, *Vijay Rahul et al. (2021)*, *Nithin, Ravula & Sulthana (2020)*, *Nur-E-Arefin (2020)*, *Prabhakara et al. (2019)*, *Pratap & Vijayaraghavulu (2021)*, *Roy & Rasheeduddin (2021)*, *Singh & Jain (2019)*, *Shekar & Ramakrisha (2021)*, *Wang & Han (2019)*. |
| | RUSBoost | 2 | *Al-Faqeh et al. (2021)*, *Arora et al. (2020)*. |
| | XGBoost (XG) | 18 | *Alias, Ibrahim & Zin (2019)*, *Almhaithawi, Jafar & Aljnidi (2020)*, *Dang et al. (2021)*, *Divakar & Chitharanjan (2019)*, *Faraj, Mahmud & Rashid (2021)*, *Gamini et al. (2021)*, *Huang, Wang & Zhang (2020)*, *Karthik et al. (2019)*, *Kolodiziev et al. (2020)*, *Layek (2020)*, *Mallidi & Zagabathuni (2021)*, *Marabad (2021)*, *Nimashini, Rathnayake & Wickramaarachchi (2021)*, *Niu, Wang & Yang (2019)*, *Parmar, Patel & Savsani (2020)*, *Trisanto et al. (2021)*, *Keskar (2020)*, *Singh, Ranjan & Tiwari (2021)* |
| | CatBoost (CB), | 3 | *Almhaithawi, Jafar & Aljnidi (2020)*, *Gamini et al. (2021)*, *Hema & Muttipati (2020)*. |
| | Gradient boosting | 12 | *Alias, Ibrahim & Zin (2019)*, *Divakar & Chitharanjan (2019)*, *Fang, Zhang & Huang (2019)*, *Faridpour & Moradi (2020)*, *Gamini et al. (2021)*, *Mallidi & Zagabathuni (2021)*, *Muaz, Jayabalan & Thiruchelvam (2020)*, *Roy & Rasheeduddin (2021)*, *Seera et al. (2021)*, *Trivedi et al. (2020)*, *Keskar (2020)*, *Taha & Malebary (2020)*. |
| | Light gradient boosted (Light GBM) | 4 | *Fang, Zhang & Huang (2019)*, *Kolodiziev et al. (2020)*, *Sellam et al. (2021)*, *Taha & Malebary (2020)*. |
| | Bagging | 5 | *Alias, Ibrahim & Zin (2019)*, *Husejinovic (2020)*, *Lin & Jiang (2021)*, *Mijwil & Salem (2020)*, *Karthik, Mishra & Reddy (2022)*. |
| | Voting | 10 | *Alenzi & Aljehane (2020)*, *Alias, Ibrahim & Zin (2019)*, *Aswathy & Samuel (2019)*, *Karthikeyan et al. (2019)*, *Kochhara & Chhabrab (2021)*, *Krishna, Nagini & Tatayyanaidu (2019)*, *Prabhakara et al. (2019)*, *Pratap & Vijayaraghavulu (2021)*, *Roy & Rasheeduddin (2021)*, *Shekar & Ramakrisha (2021)*. |
| | Pipelining | 1 | *Bagga et al. (2020)* |
| | stacking | 4 | *Karthik et al. (2019)*, *Muaz, Jayabalan & Thiruchelvam (2020)*, *Prabhakara et al. (2019)*, *Veigas, Regulagadda & Kokatnoor (2021)* |
| Deep learning | CNN | 7 | *Agarwal et al. (2021)*, *Arya & Sastry (2020)*, *Aswathy & Samuel (2019)*, *Carrasco & Sicilia-Urbán (2020)*, *Hwang & Kim (2020)*, *Ingole et al. (2021)*, *Zhang et al. (2021)*. |
| | DNN | 4 | *Arya & Sastry (2020)*, *Habibpour et al. (2021)*, *Tingfei, Guangquan & Kuihua (2020)*, *Sireesha et al. (2020)* |
| | DCNN | 4 | *Chen & Lai (2021)*, *Cheng et al. (2020)*, *Deepika & Senthil (2019)*, *Nguyen et al. (2020)*. |
| | Long short-term memory (LSTM)/BILSTM | 8 | *Agarwal et al. (2021)*, *Alghofaili, Albattah & Rassam (2020)*, *Benchaji, Douzi & El Ouahidi (2021)*, *Cheon et al. (2021)*, *Forough & Momtazi (2021)*, *Nguyen et al. (2020)*, *Osegi & Jumbo (2021)*, *Sadgali, Sael & Benabbou (2021)*. |
| | Auto-encoder (AE) | 18 | *Al-Shabi (2019)*, *Alghofaili, Albattah & Rassam (2020)*, *Arya & Sastry (2020)*, *Balne, Singh & Yada (2020)*, *Cheon et al. (2021)*, *Dzakiyullah, Pramuntadi & Fauziyyah (2021)*, *Ebiaredoh-Mienye, Esenogho & Swart (2021)*, *Huang, Wang & Zhang (2020)*, *Ingole et al. (2021)*, *Lin & Jiang (2021)*, *Misra et al. (2020)*, *Zou, Zhang & Jiang (2019)*, *Niu, Wang & Yang (2019)*, *Suthan (2021)*, *Rezapour (2019)*, *Wu, Cui & Welsch (2020)*, *Waleed, Mawlood & Jabber Abdulhussien (2020)*, *Suvarna & Kowshalya (2020)* |
| | Dual autoencoders (DAE) | 4 | *Carrasco & Sicilia-Urbán (2020)*, *Zou, Zhang & Jiang (2019)*, *Wu, Cui & Welsch (2020)*, *Waleed, Mawlood & Jabber Abdulhussien (2020)* |

(Continued)

| Learning type | Technique | Usage frequency | Reference |
|---|---|---|---|
| | Deep reinforcement learning (DLR) | 1 | *Dang et al. (2021)* |
| | Generative adversarial networks (GANs) | 7 | *Ba (2019), Fiore et al. (2019), Tingfei, Guangquan & Kuihua (2020), Hwang & Kim (2020), Niu, Wang & Yang (2019), Wu, Cui & Welsch (2020), Veigas, Regulagadda & Kokatnoor (2021).* |
| | Recurrent neural network (RNN) | 7 | *Bandyopadhyay & Dutta (2020), Chen & Lai (2021), Forough & Momtazi (2021), Hussein et al. (2021), Osegi & Jumbo (2021), Sadgali, Sael & Benabbou (2021), Zhang et al. (2021).* |
| | Gated recurrent units (GRU) | 3 | *Forough & Momtazi (2021), Sadgali, Sael & Benabbou (2021), Sadgali, Sael & Benabbou (2020).* |
| | Gradient descent algorithms | 1 | *Faridpour & Moradi (2020)* |
| | Variational automatic coding (VAE) | 1 | *Tingfei, Guangquan & Kuihua (2020)* |
| | Artificial neural network (ANN) | 36 | *Abdulsalami et al. (2019), Agarwal (2021), Akula (2020), Arun & Venkatachalapathy (2020), Asha & Suresh Kumar (2021), Daliri (2020), Deepika & Senthil (2019), Faraj, Mahmud & Rashid (2021), Forough & Momtazi (2021), Gao et al. (2019), Harwani et al. (2020), Kalid et al. (2020), Kim et al. (2019), Kumar, Mubarak & Dhanush (2020), Layek (2020), Makki et al. (2019), Makolo & Adeboye (2021), Marella et al. (2019), Mohari et al. (2020), Oumar & Augustin (2019), Muaz, Jayabalan & Thiruchelvam (2020), Nimashini, Rathnayake & Wickramaarachchi (2021), Ojugo & Nwankwo (2021), Osegi & Jumbo (2021), Varun Kumar et al. (2020), Roy & Rasheeduddin (2021), Sarvani & Markandeyulu (2021), Seera et al. (2021), Voican (2021), Uloko et al. (2021), Veigas, Regulagadda & Kokatnoor (2021), Trivedi et al. (2020), Simi (2019), Smith & Valverde (2021), Shirodkar et al. (2020), Dashora, Sharma & Bhargava (2020).* |
| | Multilayer perceptron (MLP) | 14 | *Alias, Ibrahim & Zin (2019), Arora et al. (2020), Arya & Sastry (2020), Bagga et al. (2020), Can et al. (2020), Carrasco & Sicilia-Urbán (2020), Faridpour & Moradi (2020), Hussein et al. (2021), Mallidi & Zagabathuni (2021), Misra et al. (2020), Prusti, Das & Rath (2021), Pratap & Vijayaraghavulu (2021), Seera et al. (2021), Veigas, Regulagadda & Kokatnoor (2021).* |
| | Restricted Boltzmann machine (RBM) | 3 | *Niu, Wang & Yang (2019), Suthan (2021), Suvarna & Kowshalya (2020).* |
| | Deep belief network (DBN) | 1 | *Zhang et al. (2021)* |
| Sampling technique | Synthetic minority over sampling technique (SMOTE) | 17 | *Ahirwar, Sharma & Bano (2020), Almhaithawi, Jafar & Aljnidi (2020), Dornadula & Geetha (2019), Danaa, Daabo & Abdul-Barik (2021), Dang et al. (2021), Hwang & Kim (2020), Manlangit, Azam & Shanmugam (2019), Mqadi, Naicker & Adeliyi (2021), Nguyen et al. (2020), Muaz, Jayabalan & Thiruchelvam (2020), Rtayli & Enneya (2020), Sadgali, Sael & Benabbou (2021), Sahayasakila, Aishwaryasikhakolli & Yasaswi (2019), Veigas, Regulagadda & Kokatnoor (2021), Singh, Ranjan & Tiwari (2021), Shirodkar et al. (2020), Leena & Ajeet (2019).* |
| | The adaptive synthetic (ADASYN) | 3 | *Dang et al. (2021), Vijay Rahul et al. (2021), Singh, Ranjan & Tiwari (2021)* |
| | Random oversampling (ROS) | 1 | *Singh, Ranjan & Tiwari (2021)* |
| | Tomek | 2 | *Sadgali, Sael & Benabbou (2021), Singh, Ranjan & Tiwari (2021)* |

# Appendix C

**Table C1 Comparisons of selected article on cyber fraud detection in credit card.**

| Article ID | ML/DL technique | Performance metrics | Results and value | Dataset | Future work |
|---|---|---|---|---|---|
| A1 | Back propagation neural network (BPNN). K-means | Precision, recall error rate FPR, accuracy hit and miss rate. | There is a significance difference between K-means and BPNN. BPNN model has higher accuracy comparing with K-means. BPNN accuracy = 93.1%. K-means accuracy = 79.9%. | Real credit card data/European cardholders | Comparing the effect of combing these two models together so as to optimise the accuracy. |
| A2 | LR | Accuracy, recall, precision | The model reached high performance using imbalanced dataset. L-BFGS is 0.980. Lib-linear is 0.9816. Newton-CG is 0.9812. Sag is 0.997. Saga is 0.996. | Real data/European cardholders | NA |

| Table C1 (continued) | | | | | |
|---|---|---|---|---|---|
| **Article ID** | **ML/DL technique** | **Performance metrics** | **Results and value** | **Dataset** | **Future work** |
| A3 | CNN BILSTM | Confusion matrix, accuracy, precision, recall | The proposed model (CNN-BI-LSTM-ATTENTION) achieved high accuracy in fraud detecting. Adding attention layer enhances performance. Accuracy is 95%. | IEEE-CIS fraud detection from Kaggle.com | NA |
| A4 | ANN | Accuracy Precision Recall | The ANN proposed model is best suited for detecting fraud. The accuracy around 100%. | NA | Combining this algorithm with other algorithms. |
| A5 | HMM | NA | Applying HMM model to detect credit card fraud would be successful. | NA | NA |
| A6 | RF SMOTE | Sensitivity Specificity Precision F-measure Accuracy Misclassification rate, ROC | The model showed high performance. When using RF the large number of datasets can be processed automatically. Quick RF classifier accuracy with imbalanced dataset is 98%. Quick RF Classifier accuracy with balanced dataset is 99%. | Real-world data/ UCSD FICO/2009 | NA |
| A7 | ADA boost majority balloting NB, QDA, LR, DT, RF, NN, KNN, and SVM. | -Accuracy -Matthews correlation coefficient (MCC) | Results showed that using bulk balloting technique achieves high accuracy in detecting fraud. NB: 0.9458. QDA: 0.9544. LR:0.9913 DT: 0.9837. RF: 0.9869. NN:0.971 KNN: 0.9718. SVM: 0.8526. | Genuine world MasterCard data set. | Procedures be stretched out to the internet becoming acquainted with designs. |
| A8 | K-means RF, J48 SVM | Accuracy | Results showed that RF is better on global dataset with 92.1% accuracy. K-means: 85.6%. RF: 92.1%. J48 DT: 89.3%. SVM: 89.9%. | Two types of data: Global/Bank. User dataset | For this model, the transaction time is required. |
| A9 | FraudMiner RUSBoost Bagged KNN SVM | Sensitivity False alarm rate Balanced Classification rate, MCC | This model showed great performance with catch rate 85.3% and MCC of 0.83. | Public dataset/ Provided by FISCO/UCSD | NA |
| A10 | Autoencoder LR | Confusion matrix Accuracy Recall F1-score Precision | Results showed that proposed model can detect fraud transaction between 64%, 79%, and 91%. This model is better than LR (57%) with unbalanced dataset. The model solved data balancing problem. **Balance dataset**: accuracy is 97.23. Recall is 0.90. Precision is 0.06. The F1-score is 0.12. While results on **Unbalanced dataset:** accuracy is 99.91. Recall is 0.57. Precision is 0.93 and F1-score is 0.71. | Real dataset from ULB | Compare the performance of this model with other classification algorithms. |
| A11 | LR, KNN | Confusion matrix Accuracy Sensitivity Error rate | The LR-based model is the best comparing with KNN and voting classifier. Accuracy is 97.2%. Sensitivity is 97%. Error rate is 2.8%. | Real dataset/ European cardholders | Proposed model suffers in the response time. |

(Continued)

| Article ID | ML/DL technique | Performance metrics | Results and value | Dataset | Future work |
|---|---|---|---|---|---|
| A12 | LSTM | Accuracy Loss rate Execution time | Results showed great performance of LSTM comparing with Autoencoder. Model accuracy is 99.95%. | Real dataset/ European cardholders | Calculate timing and location of fraud |
| A13 | LR, MLP, XGBoost, K-fold cross, RF, Bagging Gradient Boosting, Voting, KNN SVM, GNB. | Accuracy Confusion matrix | MLP achieved highest accuracy comparing with 15 algorithms. The accuracy is 98% | Real dataset/ European cardholders | Further research of MLP to increase the detection performance. |
| A14 | LR RF. XG CatBoost (CB), | F1-score AUC Savings | Results showed that the CatBoost obtained the best savings with 0.7158 alone. When applying SMOTE the savings is 0.971. When applying SMOTE and BMR, the saving is 0.9762. XGBoost achieved the best saving 0.757 when applying BMR without the SMOTE. XG + BMR: F1-score is (0.2890). AUC is (0.9699). Savings is (0.7570). CB + SMOTE + BMR: F1-score is (0.8250). AUC is (0.9999). Savings is (0.9762). | Real dataset/ European cardholders | Using another dataset. Also testing XG and CB |
| A15 | LR, RF KNN DT | Accuracy Precision Recall | The results show that RF achieved highest performance. RF: accuracy (95.19%), precision (0.9794), recall (0.9226). | Real dataset/ Europeans cardholders | Other data balancing techniques be explored. |
| A16 | SVM RUSBoost LR, MLP, DT, KNN, AdaBoost, RF | Accuracy Precision Specificity F1-score AUPR, ROC | The results showed that CtRUSBoost outperformed other algorithms. Results scores on three dataset: A, B, and C. **Dataset A:** sensitivity (96.30), specificity (85.60), precision (94.20), F1-score (88.60). **Dataset B:** Sensitivity (99.60), specificity (98.70), precision (95.70), F1-score (97.60). Dataset C: sensitivity 100), specificity (99.80), precision (99.30), F1-score (99.60). | Three datasets from Kaggle.com (A, B, C) | Customized the model and adding new algorithms. |
| A17 | Social spider optimisation (SSO), ant colony optimisation (ACO), ANN | Sensitivity Specificity Accuracy F-score Kappa | The model SSO-ANN achieved high performance with 93.20% accuracy on Germane dataset, and 92.82% on Kaggle dataset. | Benchmark dataset. Kaggle dataset | Improving the model by using clustering techniques. |
| A18 | Deep ensemble algorithm (DEAL). CNN. DNN. MLP, Auto encoder. SVM, LR | Mean absolute error (MAE) Fraud catching rate (FCR) Accuracy | DEAL model obtained high performance in detecting fraud. Model accuracy is 99.81% | Real dataset/ Europeans cardholders | Using AI and IoT in cloud computing |
| A19 | SVM, KNN, ANN | Confusion matrix, accuracy, precision, recall | ANN provides high accuracy in detecting fraud comparing with the unsupervised algorithms. | Real dataset/ Europeans cardholders | Na |

| Article ID | ML/DL technique | Performance metrics | Results and value | Dataset | Future work |
|---|---|---|---|---|---|
| A20 | DT IFDTC4.5 intuitionistic fuzzy logic | Accuracy, sensitivity, false positive rate, specificity | IFDTC4.5 outperforms existing techniques. The model able to detect fraud efficiently. However, still the frauds cannot be eliminated by 100%. | Singaporean bank and one similar synthetic data set. | Add multi factor authentication using the biometrics like iris, voice *etc.* |
| A21 | NB, DT, RF, CNN | Precision, Recall, Accuracy | Algorithms like NB, DT, RF and CNN are used. These algorithms are used as single models. Then these are used as hybrid models using majority voting technique. Adaptive boost also used in the model. | Publicly available credit card data set. | This model will extend to online model. |
| A22 | SVM, NB, KNN, RF | Accuracy Sensitivity Specificity Precision | Results showed that RF performs better than other algorithms. Applying sampling approach will improve the performance. NB: 97.80%. SVM: 97%. KNN: 46.98%. RF: 98.23%. | Real dataset/ European cardholders/ULB | Using huge dataset instead of sampling techniques |
| A23 | Generative adversarial networks GANs | AUC AUPRC Recall F1-score Precision | The results show that applying Wasserstein-GAN will improve detecting fraudulent transactions comparing with traditional GAN. WCGAN model achieves: AUC is 0.948. AUPRC is 0.717. Recall is 0.6420. Precision is 0.852. F1-score is 0.710. | NA | NA |
| A24 | LR, KNN, RF, NB, MLP, AdaBoost, pipeling | Accuracy Precision Recall, F1-score | The results showed that applying pipeling can improve the model's performance. Accuracy: 00.99%. Precision: 0.84. Recall: 0.86. F1-score: 0.85. | Real dataset/ European cardholders | NA |
| A25 | GNB, LR, DT, RF | Accuracy Recall Precision F1-score, MSE | The result showed that DT algorithm is the best with an accuracy: 0.999. Recall: 0.782. Precision: 0.766. F1-score: 0.774. MSE : 0.0008 | Real dataset/ European cardholders/ULB | NA |
| A26 | RNN | Accuracy, recall, precision F1-score, MSE | The result showed that RNN model is capable in detecting fraud. The accuracy is 99.87%. MSE is 0.01. F1-score is 0.99. | Synthetic dataset and real dataset | NA |
| A27 | DT, NB, SVM AdaBoost | Accuracy Sensitivity Specificity Precision ROC, F1-measure | The results showed that applying Boosting with DT outperforms other methods. The model obtained highest accuracy of 98.3%. F measure is 93.98%. Using boosting techniques improve the performance. | Real dataset/ Europeans cardholders | NA |
| A28 | DT,SVM, k-means Optimal resampling strategy, C4.5 DT AdaBoost | Accuracy Sensitivity Cost sensitive | The suggested model obtained high performance with 96.59% accuracy and 67.52% sensitivity. | Real dataset/CB bank/Brazilian bank | Compare this model with other models |

(Continued)

| Article ID | ML/DL technique | Performance metrics | Results and value | Dataset | Future work |
|---|---|---|---|---|---|
| | | | **Table C1 (continued)** | | |
| A29 | LSTM | MSE MAE, RMSE | Results showed that the LSTM model achieves perfect performance. AUC: 0.995. MSE: 0.0035. MAE:0.0065 | From the Kaggle website. | Further study of other types of RNN technique. |
| A30 | SMOTE, LOF, isolation forest, SVM, LR, DT, RF | Accuracy Precision MCC | LR, DT and RF are the best algorithms. The better parameter to deal with unbalanced data is MCC. Classifiers performing better when using SOMTE. RF: accuracy (0.9998), precision (0.9996), MCC (0.9996). DT: accuracy (0.9708), precision, (0.9814), MCC (0.9420). LR: accuracy (0.9718), precision, (0.9831), MCC (0.9438). | Real dataset/ Europeans cardholders/Kaggle | NA |
| A31 | Autoencoders | NA | The results showed that Autoencoders model most promising for detecting fraud in credit card. | Real data/European cardholders | Using balanced dataset and unhidden features. |
| A32 | NB using robust scaling | Accuracy, Precision, Recall Sensitivity AUC score F1-score | The result shows NB which used Robust Scaleris showed improvements in predicting and detecting fraud in credit card. Accuracy: 97.78%. Precision: 99.79%. Recall: 97.78. F1-score 98.71. AUC: 95.73. | Real dataset/ Europeans cardholders/Kaggle | NA |
| A33 | NB, RF, DT, MLP | Precision Recall, F-measure Specificity | The result showed that the amount-based profiling both MLP and RF obtained high improvement. This model boost fraud detection. | Dataset from 35 banks in Turkey | The high number of false positive needs further study. |
| A34 | CNN DAE MLP | Precision Recall AUC Confusion matrix ROC curves | Results showed that DNN is capable in fraud detection. MLP2OH128H918 obtained an alert reduction rate. Threshold/D (0:1) of 35.16% when capturing 91.79% fraud cases. The rate of misclassification is 8.21%. Threshold/D (0:2) of 41.47% when capturing 87.75% fraud cases. Misclassification rate is 12.25%. | Dataset from a Spanish organisation. | NA |
| A35 | DCNN, RNN, SVM, LR, RF. | Accuracy | Proposed model obtained accuracy of 99% in detecting fraud in credit card in time duration of 45 seconds. | Real dataset/ Europeans cardholders | Applying the fraud location and timing calculation. |
| A36 | 3DCNN, Spatial-temporal attention-based graph network (STAGN) | AUC Precision recall | The suggested model showed a high performance in detecting fraud in credit card. The model is effective and accurate. | Real-world data (Commercial bank) | Builds a real-time detection system. |
| A37 | Bi-LSTM-autoencoder and isolation forest | Accuracy Confusion matrix | The suggested hybrid model contains Bi-LSTM Autoencoder and the isolation forest with unbalanced data. This model obtained the highest detection rate with 87% | Real dataset/ Europeans cardholders | NA |

| Article ID | ML/DL technique | Performance metrics | Results and value | Dataset | Future work |
|---|---|---|---|---|---|
| A38 | KNN, DT, RF LR, NB | Confusion matrix recall/sensitivity precision time | Hybrid classifier/combination of supervised classifiers which worked better than any other single classifier. KNN + DT: Sensitivity: 85.63%. Precision: 86.90%. KNN + LR: Sensitivity: 57%. Precision: 85.55%. KNN + RF: Sensitivity: 82%. Precision: 95.89%. KNN + NB: Sensitivity: 58%. Precision: 80.57% | Real dataset/ Europeans cardholders/Kaggle | Use unsupervised combined classifier for batter result and use more classifier. |
| A39 | LR, RF, DT, KNN | Accuracy, specificity, precision, sensitivity | The accuracy of LR is 94.9%, DT accuracy is 91.9%, and RF accuracy is 92.9%. KNN has a 93.9% success rate. Despite LR was more accurate, majority of this algorithm under fit. Thus, KNN is the best technique. | Real dataset/ Europeans cardholders/Kaggle | NA |
| A40 | ANN Harmony search algorithm (HSA) | Accuracy, recall SM calculation confusion matrix | The suggested model NNHS provides a solution using HAS for ANN. The best accuracy achieved is 86. Recall is 87. | German dataset available at the UCI website | NA |
| A41 | HMM, SMOTE DBSCAN | Precision Recall F1-score | Proposed approach (SMOTE + DBSCAN + HMM) performed relatively better for all the various hidden states. | Simulated mobile based transactions | NA |
| A42 | Deep reinforcement. Resampling SMOTE and ADASYN | Accuracy Precision Sensitivity Specificity | The proposed model of ML with two resampling techniques and DRL is reliable. SMOTE and ADASYN are used to resampling dataset. The proposed system obtained high accuracy with 99%. RF and XGBoost are the best techniques. | Real dataset/ Europeans cardholders/Kaggle | Extend dataset. Applying new ML and DL algorithms |
| A43 | HMM | Accuracy | The model is very efficient and showed the importance in learning spending behaviour. The accuracy is 80%. | NA | NA |
| A44 | K-means. PCA T-SNE SOM | Accuracy | The model obtained an accuracy of 90%. The results were vary as the initialization of the weight of nodes SOM grid is done by randomly records or patterns. | Statlog Australian dataset. | Trying different iterations and store weights of SOM |
| A45 | KNN, RF, NB | Accuracy | KNN showed the highest accuracy than the RF algorithm and NB. | Real-world dataset | More ML supervised algorithm can be added. |
| A46 | DCNN space invariant ANN | Accuracy | The results showed that proposed robust SIANN (RSIANN) is outperformed other techniques. The accuracy is 85%. SVM accuracy is: 0.77. RF accuracy is: 0.72. NB accuracy is: 0.70. DCNN accuracy is 0.82. | NA | Using kernels technique also using pre trained CNN. |

(Continued)

| Table C1 (continued) | | | | | |
|---|---|---|---|---|---|
| Article ID | ML/DL technique | Performance metrics | Results and value | Dataset | Future work |
| A47 | SVM, NB, DT | Accuracy | Results showed that the new system will reduce the frauds which are happening while transactions. | NA | NA |
| A48 | SVM, GNB, DT | Execution times | The proposed model using fusion of detection algorithms and AI. Support Vector Classifier take less time. SVC obtained solution with less time. 0.191343 ms. | Real data/European cardholders | Using other datasets also applying other algorithms |
| A49 | RF | Accuracy | The result showed that RF obtained high performance. However, the speed will suffer. On the other hand, SVM suffer from unbalanced data. The SVM obtained good performance. | NA | NA |
| A50 | NB, DT, RF, LR AdaBoost Gradient Boost XGBoost | Accuracy Recall Precision Confusion matrix | Results showed that XGBoost is the best boosting technique in predicting fraud. The accuracy is 100%. F1-score is 0.88. NB classifier: 95.6%. DT classifier: 90.0%. RF classifier: 97.7%. LR: 98.3%. AdaBoost: 99.9%. Gradient boost: 99.9%. XGBoost: 100%. | Real dataset/ Europeans cardholders/Kaggle | NA |
| A51 | RF, DT LR, LOF Isolation forest | F1-scores Precision Recall | Results showed that isolation forest obtained better efficiency. RF: 95.5%. DT: 94.3%. LR: 90%. Isolation forest: 99.77%. Local outlier factor: 99.69%. | Real dataset/ Europeans cardholders | Using NN for training the system, to obtain better accuracy. |
| A52 | Semi-supervised learning. AutoEncoders | Precision Recall F1-score | The results show that using semi-supervised technique is efficient to detect fraud. Accuracy is 0.98%. | Real dataset/ Europeans cardholders | Investigate the intelligent dependent attributes. |
| A53 | Autoencoders | Accuracy Precision F1-score Sensitivity | The proposed model obtained high performance. SSAE+LDA model showed significant improvement comparing with other research on same dataset. Accuracy is 90%, F1-score is 90%, precision is 91%, sensitivity is 90%. | Real dataset/UCI | Study effect of optimizers, stacking diverse autoencoders |
| A54 | Light gradient boosting. RF | Accuracy AUC | This study only used to identify the fraudulent user. The results show that light gradient boosting obtained great performance with a total recall rate of 99%. | Real dataset/ Europeans cardholders | Further study on how to judge fraud ring based on relation map. |
| A55 | XGBoosting Neural network | Accuracy Precision, F1-score Recall, ROC, AUC | Results indicated that XGBoosting performs better when comparing with other ensemble models. XGB AUS is 0.778 | Consumer's dataset/ Taiwan. | NA |
| A56 | MLP LR, SVM. Gradient descent algorithms. | Accuracy | Results showed that proposed model performs good comparing with LR and SVM. MLP Accuracy: 0.9990 LR Accuracy: 0.9723 SVM Accuracy: 0.9345 | NA | A dependent variable with numerous classifications can be used. |

| Article ID | ML/DL technique | Performance metrics | Results and value | Dataset | Future work |
|---|---|---|---|---|---|
| A57 | GAN | Accuracy Precision | The model obtained an improved sensitivity. GAN model can training of small dataset. GAN Accuracy: 0.99962. Precision: 0.9583. | Real dataset/ Europeans cardholders | Develop a strategy to reduce the decreasing in specificity to minimum |
| A58 | Ensemble learning approach RNN, FFNN LSTM, GRU | Recall Precision F1-score | Results showed that proposed model based on LSTM with ensemble GRU on two datasets outperforms other models. The new model is efficient in term of realtime. | -Real dataset/ Europeans cardholders. -Brazilian bank | Develop new Model to take advantage of deep encoder and decoder. |
| A59 | CatBoost XGBoost Stochastic gradient boosting | Precision Recall Confusion matrix | Results showed that the CatBoost is the best comparing with XGBoost and SGB boosting. CatBoost accuracy is 0.921. Recall is 1.00. XGBoost accuracy is 0.914. Recall is 0.99. SGB accuracy is 0.907. Recall is 0.97. | NA | New models using supervised and unsupervised. |
| A60 | LR, ANN SVM, RF Boosted Tree | Kolmogorov-Smirnov Formula. FDR | The new model using boosted tree shows best performance in fraud detection. FDR = 49.83% | Real dataset/ government agency/USA | Some data and fields such as time, day point of sale should be added. |
| A61 | NB, RF LR, SVM | AUC Precision Recall | NB technique shows high performance comparing with other techniques. Accuracy is 80.4%. Area under curve is 96.3% | Real dataset/ Europeans cardholders | Develop another model for sampling imbalanced data. |
| A62 | Isolation forest | Precision-recall curve (AUCPR) AUC | The proposed model demonstrate the efficiency in fraud detection, observed to be 98.72%, which indicates a significantly better approach than other techniques. | Real dataset/ Europeans cardholders/Kaggle | Financial institutions must make available data set. Thus, outcome will be more efficient. |
| A63 | UQ techniques: MCD EMCD | Confusion matrix UAcc, USen USpe, UPre | The suggested model using UQ provide high performance in predicting fraud. Ensemble technique is efficient in fraud prediction. MCD: UAcc (0.82) Ensemble: UAcc (0.85). EMCD: UAcc (0.84) | Publicly available dataset/Vesta corporation | The quality of final uncertainty estimates should be improved. |
| A64 | RF, LR, DT, GNB combination with ensemble. | Matthews correlation coefficient (MCC) | The accuracy of all the five models is 100% & even the MCC score is +1 for the models been evaluated. | Real dataset/ Europeans cardholders | NA |
| A65 | DT augmented with regression analysis. | Accuracy Confusion matrix | The results showed that new model successfully verified the injected intrusions. Accuracy is 81.6% with 18.4% misclassification error. | Dataset from the UCI repository | NA |
| A66 | SOM ANN | Accuracy | Using hybrid model of SOM and ANN achieved high performance compared to use ANN or SOM alone. | Dataset from the UCI repository | Creating a NN with some optimization technique. |

(Continued)

| Article ID | ML/DL technique | Performance metrics | Results and value | Dataset | Future work |
|---|---|---|---|---|---|
| A67 | LR, RF, and CatBoost | Accuracy Precision Recall | The result showed that model of RF with CatBoost provides efficient accuracy. RF technique has the most elevated incentive than the LR and CatBoost algorithm.. RF: Accuracy (99.95). CatBoost: Accuracy (99.93). LR: Accuracy (99.88). | Real dataset/ Europeans cardholders/Kaggle | NA |
| A68 | Deep forest XGBoost AE, gcForest | Accuracy Precision, Recall Confusion matrix | The proposed model showed high performance in detecting card fraud. | Dataset from China's bank. | NA |
| A69 | GAN, variational automatic coding (VAE) | Accuracy F-measure Precision | The model showed that VAE-based oversampling performs better than the normal DNN and synthetic minority over sampling technique as it can solve the imbalanced problem. | Real dataset/ Europeans cardholders | Improving the model recall rate |
| A70 | C4.5 DT, NB Bagging ensemble | Accuracy Precision, Recall | The model shows that bagging with C4.5 DT is the best algorithm with rate of 1,000 for class 0.0825 for class 1. | Real dataset/ Europeans cardholders | NA |
| A71 | Fuzzy rough nearest neighbor (FRNN) SMO, LR, MLP, NB, IBK, RF | Positive predictive value (PPV). F-measure Specificity PPV, F-measure | The results showed that the suggested model provided significant results. The rate of detection is 84.90, AUC is 0.8555/Australian dataset. While 76.30% detection rate with 0.679 AUC/German dataset. | Australian dataset/ German dataset | Other ensemble techniques should be considered. |
| A72 | NB, DT, (LMT, J48,) Rules classifier Lazy classifier Meta classifier | Accuracy Recall Precision F1-score | The result showed that applying LMT algorithm to classification fraud is better than other techniques. LMT model obtained 82.08% accuracy. | Client's data in Taiwan. Data available on: https://archive.ics.uci.edu. | Further study to find out new algorithms with higher voting. |
| A73 | Feature maps and GANs SVM CNN | AUC score ROC Confusion matrix | Results showed that the suggested model is applicable to test datasets and less time is required for learning. SVM obtained better detection. However, learning time exceeds other models when dataset increase. CNN-based model needs long time. SOMTE performance is effective. | Machine learning group ULB. Kaggle. | Change on oversampling techniques in the suggested model. |
| A74 | CNN,SVM, RF isolation forest Autoencoder | Accuracy Precision | ML models have been implemented for classification purpose. Achieved competitive accuracy in CNN model. CNN: Accuracy (99.51). | Real dataset/ Europeans cardholders | Predict fraud in real-time. Applying service on the cloud platform. |
| A75 | LR, NB, KNN | Accuracy, Recall Specificity Sensitivity F-measure Precision | Results showed that LR showed optimal performance. It is getting high accuracy of 95%. NB accuracy is 91%. KNN accuracy is 75%. LR showed better sensitivity, precision, specificity, and F-measure. | Real dataset/ Europeans cardholders/Kaggle | NA |

| Article ID | ML/DL technique | Performance metrics | Results and value | Dataset | Future work |
|---|---|---|---|---|---|
| A76 | Isolate forest and local outlier factor (LOF) algorithms | Accuracy Precision Recall F-measure | The result showed that local outlier factor achieved high accuracy with 97%. Isolation forest accuracy is 76% | Real dataset/ Europeans cardholders/Kaggle | NA |
| A77 | RF, DT | Accuracy Sensitivity Specificity Precision | The result showed that this model is accurate on large dataset with 98.6% accuracy. RF provides high performance, however, it needs many training data. | Dataset from product reviews on credit card transaction. | Develop AI/ML/DL techniques |
| A78 | Multiple classifiers system (MCS). NB, C4.5, KNN, ANN, SVM. | TPR TNR Accuracy | Results showed that the suggested model can tackle the unbalanced class distribution and overlapping class samples. The proposed model obtained high TPR, which is 0.840 and 0.930 accuracy. TNR is 0.955. | Dataset1: ULB Dataset2: credit cardholders/ Taiwan bank | Considering combining the DL algorithms for promising detection results. |
| A79 | KNN, SVM, LR HYBRID NB-RF XGB | Accuracy, recall, precision, TPR, FPR, | Results showed that all proposed models are superior in performance. Staking classifier using LR as meta classifier is most promising then SVM, LR, KNN and HNB-RF. Stacking classifier accuracy is 0.95. RF accuracy is 0.94. | Real dataset/ Europeans cardholders/Kaggle | Applying Voting classifier. |
| A80 | Hybrid models using AdaBoost and majority voting, NB, SVM | MCC | Results showed that the majority voting obtained high accuracy. The best MCC score is 0.823. | A publicly available data set/Turkish bank. | Applying online learning models so we enable efficient fraud detection. |
| A81 | DT, LR, Shallow NN. Challenger model: DL model with ensemble. | AUROC K–S statistics alert rate, recall precision | Results showed that after testing off-line and post-line, operate the FDS with DL model. This shown +3.8% improvement of recall. The hybrid ensemble model perform well in detecting fraud. | Dataset from company/South Korea | NA |
| A82 | LR, NB, AdaBoost, and voting classifier | Accuracy, recall, precision, sensitivity F1-score | Results showed a good accuracy for NB: 91.41%. LR: 94.51%. AdaBoost: 95.67%. Voting: 94.69%. | Real dataset/ Europeans cardholders/Kaggle | A hybrid classification method will be designed. |
| A83 | Ensembles of classifiers based on DT, XGBoost and LightGBM. | Accuracy, precision, recall AUC Confusion matrix | The result showed that the ensemble of models allowed to detect maximum 85.7% of fraud. Accuracy is 79–85%. | Real dataset/ Europeans cardholders/Kaggle | NA |
| A84 | AdaBoost voting KNN, greater part casting ballot techniques. | MCC | The results showed that perfect MCC score achieved when using AdaBoost and greater part casting a ballot. Commotion from 10% to 30% included with data. The model yielded best MCC of 0.942. | Informational index from a Turkish bank | NA |
| A85 | RF, KNN, NB, SVM | Accuracy | The result shows that RF has the highest accuracy of detection of fraud. RF accuracy is: 0.9996. | NA | Seeking information from advanced technologies. |

(Continued)

| Article ID | ML/DL technique | Performance metrics | Results and value | Dataset | Future work |
|---|---|---|---|---|---|
| A86 | ANN, BBN | Confusion matrix | Result showed a Bayesian Network is more accurate than the NB Classifier. This is disturbed with using the fact of conditional dependence between the attributes in Bayesian Network, but it requires more difficult to calculation and as training process. | Real dataset/ Europeans cardholders/Kaggle | NA |
| A87 | KNN, NB, LR | Accuracy, sensitivity, specificity, | The result showed that KNN performed high performance of matrices except accuracy. | Real data/European cardholders | NA |
| A88 | LR, RF, SVM | Accuracy, precision, F1-score, recall | Compression between LR, RF and SVM is performed and the accuracy of LR is 77.97%, RF is 81.79% and SVM is 65.16. So, RF is better than the SVM and LR. | Real dataset/UCI | NA |
| A89 | LR, RF, XGBoost, ANN, isolation forest, PCA with SVM. | Accuracy, sensitivity, specificity, MCC precision, BCR | Results show that RF and XGBoost provided better result than other models. The accuracy of XGBoost is 0.9951. RF accuracy is 0.9955. | Mobile money transactions published on Kaggle. | Combined ANN with genetic algorithm to enhance accuracy. |
| A90 | SVM, GA Cuckoo search Particle swarm | Accuracy Precision Recall | The results showed that Linear kernel function is the best. Accuracy is 91.56%. Radial basis used to enhance kernel accuracy. The accuracy improved from 42.86 to 98.05%. Overall, PSO-SVM better than CS-SVM and GA-SVM. | Data from law enforcement department in China | Look for new algorithms to optimize SVM |
| A91 | AE-PRE Bootstrap aggregating Bagging | Accuracy TPR, TNR FPR ROC curve AUC, MCC | The result shows that AE-PRF is efficient when dataset is unbalanced. AE-PRF obtained high performance in accuracy. | Real dataset/ Europeans cardholders/Kaggle | Improve AE-PRF model with adding fine-tuning the hyperparameters of AE and RF models. |
| A92 | Multi-perspective HMMs | PR-AUC | The results showed that HMM model is powerful in detecting fraud. | Real dataset/Belgian | Combine LSTM with HMM-base features |
| A93 | C5.0, SVM, ANN NB, BBN, LR, KNN, artificial immune systems (AIS). | Accuracy Recall Precision | The results showed that C5.0, SVM, and ANN are performing well with imbalanced classification problem. Even these techniques improve the classifier's performance in fraud, high number of fraud cases continue undetected. | Two dataset available at http://packages.revolutionanalytics.com/datasets | Develop new model with big data driven ecosystem. |
| A94 | Hybrid model: DT, SVM, ANN genetic algorithm (GA). | F-score Accuracy Recall | The results showed that the suggested hybrid model obtained high accuracy with 93.5% comparing with ANN, SVM, and DT. The hybrid model applied GA outperform other techniques. | Realworld dataset from financial institution | Real-life test for the suggested model |
| A95 | DT, RF, KNN, LR K-means, DBSCAN, MLP, NB, XGBoost Gradient boost | Accuracy Precision Recall F1-score | The result showed that RF yielded perfect performance result with accuracy 99.995. RF is suitable for large datasets. | Real dataset/ Europeans cardholders/Kaggle | NA |

| Article ID | ML/DL technique | Performance metrics | Results and value | Dataset | Future work |
|---|---|---|---|---|---|
| A96 | Local outlier factor. Isolation forest | Precision Accuracy | The results showed that the model reached over than 99.6% accuracy. Precision at 28%. When fed more data in the model, the precision raised to 33%. | Dataset from German bank in 2006. | Adding more algorithms. Using more dataset. |
| A97 | KNN, PCA, SMOTE | Recall Precision F1-score | The results showed that the suggested model performed well. For KNN: Precision 98.32. F-score 97.44%. For Time subset when using the misclassified instance, precision is 100% and F-score is 98.24%. | Real dataset/ Europeans cardholders/Kaggle | Know how PCA can affect the performance of a dataset. |
| A98 | KNN, DT, LR RF, XGBoost | Accuracy F1-score Precision Recall, AUC-ROC | The results show that the XGBoost and DT outruns all other algorithms in detecting fraud. | Real dataset/ Europeans cardholders/Kaggle | Study on other ML algorithms and various forms of stacked classifiers. |
| A99 | Outlier detection DT, RF and NN | Precision Recall ROC Confusion matrix | The results showed that RF is the most precise and accurate technique. However, it takes long time to train. NN is the next best algorithm. DT is the least accurate. In term of time efficiency and computational resource utilization the NN is the best technique. | Real dataset/ Europeans cardholders/Kaggle | NA |
| A100 | NB, C4.5 DT, and bagging ensemble learner. | Precision Recall PRC | The result showed that the performance is between 99.9% and 100%. The best classifier is C4.5 DT with 94.1% precision and 78.9% recall. The acceptable performance is bagging ensemble with 91.6% precision and 80.7% recall. As for the worst performance, it is the NB classifier with precision of 65.6% and a recall of 81%. | Real dataset/ Europeans cardholders/Kaggle | Other classifiers will be used and applied to a set of local data that will be collected from banks in Iraq. |
| A101 | Autoencoders MLP, KNN and LR | Accuracy Precision Recall F1-score | Results showed that the suggested model maintains a good performance. It outperforms the systems based on either different classifiers or variants of autoencoder. It establishes the efficiency of proposed two stage model. Proposed method accuracy is 0.9994. Precision is 0.8534. F1-score is 0.8265. | Dataset from ULB machine learning group on Kaggle. | Proposed two stage model can be tuned to handle stream data. The model can be trained on a batch of transactions. |
| A102 | LOF, AdaBoost, RF, isolation forest, DT, KNN, HMM, GA, ANN, NB, LR | Accuracy Confusion matrix | Results showed that the local outlier factor accuracy is greater than other algorithms. Local outlier factor accuracy is: 0.898. | Real data/European cardholders | NA |

*(Continued)*

| Article ID | ML/DL technique | Performance metrics | Results and value | Dataset | Future work |
|---|---|---|---|---|---|
| A103 | RF | Accuracy | The results showed that RF performs better with large dataset. The accuracy is 99.9%. The SVM algorithm can be used instead of RF. However, SVM still suffers from the imbalanced dataset. | NA | Privacy preserving techniques can be applied in distributed environment. |
| A104 | RF | Accuracy F1-score, Precision, Recall | The result showed that the RF performed better comparing with DT and NB. The suggested model showed better accuracy on huge dataset. | Real dataset/100,000 cardholders | Applying semi-supervised technique |
| A105 | Oversampling with SMOTE SVM, LR, DT, RF | Accuracy Precision Recall, F1-score | Results showed that when using SMOTE technique, the model works better in predicting fraudulent. RF and DT provided best performance. | Real dataset/ Europeans cardholders | Building a real-time solution to detect fraud. |
| A106 | RF AdaBoost oversampling ADASYN | Accuracy Recall Precision F1-score | This research examines various existing credit card fraud systems using ML approaches. Despite the fact that RF produces outstanding results on tiny sets of data, there are still certain problems, such as data imbalance. RF accuracy is: 0.999. | Real dataset/ Europeans cardholders/Kaggle | Using large amount of data. More pre-processing procedures. |
| A107 | Autoencoder neural network DAE | Recall Accuracy | The results showed the DAE improves classification accuracy of minority class of imbalanced datasets. Proposed model increases accuracy of minority class. When threshold equal to 0.6, model achieves best performance with 97.93%. | Real dataset/ Europeans cardholders/Kaggle | Dimensionality reduction of high-dimensional data needs further research. |
| A108 | ANN with LR | Accuracy, Precision and Recall | The results show that the model is very good. Accuracy achieved of 0.9948, the recall is 0.8639 and precision of 0.2134. | Real data/European cardholders | NA |
| A109 | GAORF | Accuracy Confusion matrix | The results showed that using real and genetic algorithm optimised RF models. The model has good improvement and bringing down misclassifications. | Commercial bank in Nigeria | NA |
| A110 | 2DCNN, 1DCNN LSTM, NLP SMOTE | Accuracy F-score Precision Recall | The result showed that using CNN and LSTM yielded better performance. LSTM (50 blocks) was the highest with F1-score of 84.85%. Sampling techniques applied to solve imbalanced dataset and improve model performance. | Real dataset/ Europeans cardholders/Kaggle | Hyperparameters to build DL techniques to improve performance. |

| Article ID | ML/DL technique | Performance metrics | Results and value | Dataset | Future work |
|---|---|---|---|---|---|
| A111 | ANN, RF, GBM RUS, SMOTE DBSMOTE SMOTEENN | F1-score Recall Precision Accuracy | The result showed that using sampling techniques enhanced the detecting of fraud in credit card. Recall obtained with SMOTE by DRF classifier is 0.81 which is the best. Precision is 0.86. Staked ensemble shown promise in detecting fraud. | Real dataset/ Europeans cardholders/Kaggle | Using other sampling techniques. Applying unsupervised and semi-supervised techniques. |
| A112 | Local outlier factor, LR, RF, DT isolation forest | Accuracy MCC | The result showed that the LR, SVM obtained higher accuracy. SVM accuracy is 0.9987. LR accuracy is 0.9990. One-class SVM applied in this study. | Real dataset/ Europeans cardholders | NA |
| A113 | ANN.LR, DT, RF and XGBoost | Accuracy Precision Recall F1-score | Results showed that ANN and XGBoost performed a high performance. ANN achieved a 99% accuracy. | -Real dataset/ Europeans cardholders. -Synthetic dataset | Use more real world datasets. |
| A114 | NB, SVM AdaBoost | MCC Accuracy | The results showed that boosting technique achieved a good accuracy. The best MCC score is 0.823. | Real world dataset. | Extend the model to online learning model. |
| A115 | KNN, LR, SVM, RF, DT, XGB, OCSVM, AE, RBM, GAN | AUROC FPR TPR | The results showed that applying supervised approach such as, RF and XGB achieved better performance. XGB obtained 0.989 AUROC. RF obtained 0.988 AUROC. Unsupervised techniques RBM achieved the best performance with 0.961 AUROC. | Real dataset/ Europeans cardholders/Kaggle | Focuses on new GAN model |
| A116 | RF | Accuracy Sensitivity Specificity Precision | The result showed that building multiple DT achieved good performance with 98.6% accuracy. | Real dataset/ Europeans cardholders/Kaggle | NA |
| A117 | IBk, IB1, KStar, RandomCommittee, and RandomTree AdaBoost | Accuracy Precision Recall | The results showed that the best accuracy achieved by Bagging, Rotation Forest, Random SubSpace, Random Committee, LMT, and REPTree. The IBK, IB1, RandomCommittee, KStar, and RandomTree obtained good accuracy. And can detect fraud 348 (35.27%), 354 (40.97%), 396 (45.83%), 397 (45.94%), and 399 (46.18%) respectively. | UCSD—FICO dataset. | NA |
| A118 | Spectral-clustering hybrid of GA trained modular NN. | Sensitivity Specificity Accuracy | Results showed that hybrid model is efficient in detecting fraud. The model obtained sensitivity of 90%, specificity of 19% and prediction accuracy of 74% with improvement rate of 12% for data inclusion. | Dataset from banks/ Africa and Nigeria. | NA |

*(Continued)*

| Table C1 (continued) | | | | | |
|---|---|---|---|---|---|
| Article ID | ML/DL technique | Performance metrics | Results and value | Dataset | Future work |
| A119 | ANN, SA-ANN HTM-CLA DRNN, LSTM | Accuracy | Results showed that the HTM-CLA offered a realistic features. HTM-CLA with SA-ANN achieved good performance. The maximum accuracy obtained from SA-ANN. | Real dataset/ Australia Real dataset/ German | Reduce computational burden in HTM-CLA technique |
| A120 | Isolation forest KNN, DT, LR, RF | Sensitivity time and precision | The result showed that KNN sensitivity is better than DT. However, DT needs less time to detect fraud. DT is the best model. | Real data/European cardholders | NA |
| A121 | LR, DT, RF, NB, ANN | Accuracy Recall Precision | Results showed that the accuracy is 94.84% when using LR. 91.62% when using NB and 92.88% when using DT. ANN obtained better accuracy of 98.69%. ANN is the best. | Real dataset/ Europeans cardholders | NA |
| A122 | KNN, DT, SVM, LR, RF XGBoost | Accuracy F1-score Confusion matrix | The result showed that KNN model is the best comparing with other techniques. Accuracy is 99.95%. F1-score is 85.71%. | Real dataset/ Europeans cardholders | Using other resampling and applying DL techniques. |
| A123 | Hybrid architecture involving the optimization of the particles swarm (PSO) SVM | Accuracy F1-score Confusion matrix | The PSO algorithm is used to select characteristics and the SVM is used for the iterative development of the feature selection. Results shown that a minimum of functionalities is extracted by the suggested PSOSVM. The PSO-SVM algorithm is an optimal preparatory instrument for enhancing feature selection optimisation. Accuracy for German dataset: with SVM: 78.69. PSOSVM: 89.42. Accuracy for Australian Dataset: with SVM: 78.84.PSOSVM: 89.27. | German credit card datasets. Australian credit cards | NA |
| A124 | Stacking AdaBoost majority voting LR, DT, RF | Accuracy | The result showed that the suggested model provided better fraud detection. The boosted stacking performs better than others. Boosted Staking accuracy is 94.5% | Real dataset/ Europeans cardholders/Kaggle | NA |
| A125 | Neo4j, PageRank, RF, DT KNN, SVM, MLP, LOF, isolation forest | Accuracy MCC, F1-score Recall, Precision ROC, AUC, AUPR | The result showed that significant improvement in performance metrics of DT. LOF yielded a better result with 99.54% accuracy and recall 83.39%. When using PageRank graph feature. RF accuracy is 99.47%. | Synthetic dataset/ BankSim | Other graph algorithms to extract feature and DL should be studied further. |
| A126 | Autoencoder, RBM | Recall, Precision AUC | The result showed the AE and RBM can make AUC more accurate. AE based camera and H2O applied. | Real dataset/ Europeans cardholders | NA |

| Article ID | ML/DL technique | Performance metrics | Results and value | Dataset | Future work |
|---|---|---|---|---|---|
| A127 | AdaBoost Majority vote MLP, SVM LOR, HS | MCC metrics | The result showed that the hybrid model of majority voting provided good accuracy. The model achieved great location rate 98% with 0.1%. Perfect MCC score when using AdaBoost and Majority voting. | Real dataset/ Europeans cardholders/Kaggle | Examined other internet study models |
| A128 | AE, one-class SVM and robust Mahalanobis outlier detection | Precision Error rate MSE | Results showed that the advantage of robust Mahalanobis is that does not need label for training. The performance of the three models was vary. To get vision about performance of models the available labels used for model performance evaluations. | Real dataset from international corporation | Global and local outlier, cardholder behaviour need to be considered. |
| A129 | AdaBoost, NB, RT Majority voting DT, GBM, NN, SVM, Spark ML | MCC | The results showed that the hybrid model of NB, SVM, and DL techniques obtained an ideal MCC score 0.823. | Public real dataset/ bank | Expand to internet learning |
| A130 | SVM-RFE Hyper-parameters Optimization SMOTE | Accuracy Precision Recall, Specificity F-score | Results showed that the proposed model is high effective and obtained the best accuracy with 99%. | Real dataset/ Europeans cardholders/Kaggle | Using more complex datasets |
| A131 | RNN SMOTE Tomek LSTM, BLSTM GRU, BGRU | Accuracy Recall, Precision AUC | Results showed that BGRU achieved the best accuracy 97.16%, then BLSTM with 96.04%. | Real dataset/ Europeans cardholders/Kaggle | Focuses on the behavior of customer. |
| A132 | WOA SMOTE BPNN | Accuracy | The result showed that the WOA and SMOTE obtained more efficient than BPNN. | Real dataset/ Europeans cardholders | NA |
| A133 | NB, SVM, RF | Accuracy | The results showed that the RF is the best technique with accuracy of 100%. | Real data/European cardholders | NA |
| A134 | RF, SVM, LOF isolation forest | Accuracy Precision Recall | The result showed that the RF obtained 99.92 accuracy. RF performed better comparing with other techniques. | Real dataset/ Europeans cardholders | Improve dataset and add other algorithms to the suggested model |
| A135 | K-means C5.0 DT Hadoop and Spark | Accuracy ROC AUC | The results showed that the spark-based IHA hybrid model obtained 94% accuracy. It is suitable for detect fraud. | Public domain http://packages. revolutionanalytics. com/datasets | Applying this model to other fields |
| A136 | SVM Undersampling techniques | Accuracy Precision Recall | Results showed that the new model improves the performance. Accuracy is 99.9%. SVM obtained best precision with 89.5%. | Real dataset/ Europeans cardholders | NA |
| A137 | Isolation forest | Accuracy | The results showed that the isolation forest obtained accuracy with 99.87. | Professional survey organizations. | Using hybrid techniques and AI |
| A138 | RF | Accuracy | The results showed that RF using feedback and delayed supervised sample is better than other techniques. RF accuracy is 0.962. | NA | Applying semi-supervised techniques |

(Continued)

| Article ID | ML/DL technique | Performance metrics | Results and value | Dataset | Future work |
|---|---|---|---|---|---|
| A139 | SVM, KNN AdaBoost PSOS RIG | Accuracy Precision Recall F-measure | The results point out that PSOS technique is the best feature optimisation technique. This technique enhanced the accuracy from 82.90% to 85.51%. PSOS technique gives more performance. | Australian financial dataset. | Extend the model by using hybrid techniques |
| A140 | AdaBoost majority voting, NB, SVM, DL | MCC | The results showed that Majority voting obtained a high accuracy and best MCC score with 0.823. | Public realworld data set | Extend to online learning model |
| A141 | RF, NN | Accuracy Precision Recall, F-measure | The result showed that RF obtained accuracy with 90%. RF is suitable technique. | Real-life B2C dataset | Thr RF itself needs improvement. |
| A142 | NB, RF, DT, GBT, DS, ANN, RT, MLP, LIR, LOR, SVM | Accuracy ACC MCC | The results showed that the best AUC obtained is 0.937 from GBT using aggregated features. Aggregated features improve the models performance. | Public data sets. Benchmark databases. | Further evaluation of this models using different datasets. |
| A143 | HOBA DBN, RNN, CNN BPNN, SVM, RF | Accuracy Precision, Recall F1-measure | The results showed that the DBN with HOBA variable obtained better performance. Using DL techniques and HOBA feature engineering improve the performance. | Real-world dataset/ bank in China | Build real-time model. Build a combination model of ML and DL |
| A144 | AE GAN | Precision Recall F1-measure | The result shows that the DAEGAN model achieved best performance. AUC is 0.958. Recall is 0.815. AUPRC is 0.805. DAEGAN improves accuracy. | Real dataset/ Europeans cardholders | Improve the model |
| A145 | Isolation forest | AUCPR F1-score Precision, Recall ROC-AUC | The result showed that the model achieved good performance. AUCPR is better than ROC-AUC in describing performance. Precision is 0.807. Recall 0.763. F1-score is 0.784. ROC-AUC is 0.973. AUCPR is 0.759. | Real-life dataset from ULB. Kaggle. | NA |
| A146 | SVM, LOF isolation forest SVM | Accuracy Precision F1-score, Recall | The results point out that isolation forest with LOF model very fast and accurate. The accuracy is 99.74%, SVM obtained 45.84%. LOF achieved 99.66%. | NA | NA |
| A147 | SVM, K-means AdaBoost | Recall, Accuracy | The result showed that SVM and AdaBoost obtained high performance. | Dataset from a bank. | NA |
| A148 | Deep auto-encoder | Accuracy Precision, Recall AUC-ROC Curve | The results showed that the algorithm is perfect and gave high performance 98.8% acceptance rate. The proposed algorithm can be used for any Binary classification task. | Real dataset/ Europeans cardholders | NA |

| | Table C1 (continued) | | | | |
|---|---|---|---|---|---|
| **Article ID** | **ML/DL technique** | **Performance metrics** | **Results and value** | **Dataset** | **Future work** |
| A149 | NN | Accuracy | The result showed that the suggested model can be integrated with mobile apps to detect fraud. Model obtained excellent accuracy with 99.75%. | Real dataset/ Europeans cardholders | NA |
| A150 | RF, DT, SVM, GNB LR | Accuracy | The result showed that the DT provided better performance. However, speed still suffer. | NA | Using other ML and DL techniques |
| A151 | Isolation forest LOF | Recall, Precision F1-score | The isolation forest obtained accuracy with 99.72%. With number of errors 71. LOF accuracy is 99.62% and number of errors 107. Isolation forest is better in detecting fraud. | Real dataset/ Europeans cardholders | Using NN technique |
| A152 | DT, RF, HMM, NN | Accuracy false alarm rate, MCC | The results point out that the RF obtained high performance with 0.999% accuracy in fraud detection. | Real dataset/UCI | NA |
| A153 | TVIWDA SVM WFSVM | Accuracy Precision, Recall F1-score | The result showed that using TVIWDA with WFSVM improved the accuracy of detection. The suggested system obtained 97.82% accuracy. Precision is 92.62%. | German credit card dataset. | Solving the imbalanced data problem |
| A154 | Oversampling pre-processing technique SAS. RF, KNN, DT, LR | Accuracy | The study proposed 4 models to detect credit card fraud. The result showed that the RF and KNN are overfitting. Thus, only the DT and LR have been compared. The best performing model is a LR. Result shows that LR with stepwise splitting rules has outperformed the DT with only 0.6% error rate. | Real dataset/ Europeans cardholders/Kaggle | Use different sampling technique such as undersampling, SMOTE or roughly balancing to compare the result. |
| A155 | RF, DNN | Accuracy | The results showed that the RF perform perfect with large number of data. RF accuracy is 0.999. | NA | NA |
| A156 | KNN, LR | Accuracy Precision Recall, F-measure | The result shows that the KNN technique is achieved best result. Precision is 0.95. Recall is 0.72. F1-score is 0.82. | Real dataset/ Europeans cardholders | NA |
| A157 | SMOTE MLP, KNN, SVM OSE, NN, GAN | Accuracy F1-score | The results point out that the model using stacking classifier which combines GAN-improved MLP with SVM and KNN. OSE is preferred because of its ability to harness the abilities of MLP which works better in finding hidden patterns. The accuracy of OSE is 99.8% | Real dataset/ Europeans cardholders/Kaggle | Apply weighted voting and boosting algorithms |

(Continued)

| Article ID | ML/DL technique | Performance metrics | Results and value | Dataset | Future work |
|---|---|---|---|---|---|
| A158 | Aggrandized RF | Accuracy Precision Recall, F-measure Sensitivity Specificity | The result showed that the aggrandized random forest is obtained high accuracy with 0.9972% for balanced data. And 0.9995% for imbalanced data. RF is the best technique in detecting fraud. | NA | NA |
| A159 | RF, ANN, SVM, LR, tree classifier gradient boosting | Accuracy Precision Recall F1-score | Results showed that the RF algorithm demonstrate an accuracy percentage with 95.988%. SVM accuracy is 93.228%. LR accuracy is 92.89%. NB accuracy is 91.2%. DT accuracy is 90.9%. GBM accuracy is 93.99%. | ULB dataset from Kaggle | Apply other ML techniques |
| A160 | SVM, NB, KNN focal loss XGBoost W-CEL, LR | Accuracy Precision Recall MCC | The result showed that the suggested model achieved accuracy with 100%. Precision is 0.97. Recall is 0.56. MC is 0.72 using extreme imbalanced dataset. When using mild balanced dataset, the accuracy is 99%. 0.88 precision. 0.87 recall. 0.89 MCC. The suggested model is not working well when using extreme dataset. XGBoost improves model performance. | ULB dataset from Kaggle | Solve the imbalanced dataset problem |
| A161 | HMM | NA | The study provided a method to find out the spending behaviour of cardholder, then find out the observation symbols so that help in estimating the model performance. | NA | NA |
| A162 | LOF, K-means isolation forest | Precision Recall F1-score | The result shows that proposed model provided an accuracy with 98%. K-means clustering, isolation forest and LOF. | Real dataset/ Europeans cardholders | NA |
| A163 | KNN, LR, RF XGBoost extreme gradient boost | Precision ROC-AUC | As the XGBoost is showing more accuracy than other models. Out of these algorithms, XGBoost model is preferable over the RF model and LR model. | Real data/European cardholders | RF model would be improved |
| A164 | AdaSyn, ROS, RUS, Tomeklinks AIIKNN, Tomek SMOTE+ENN, AdaBoost, KNN, RF, SVM, eXtreme XGBoost, LR | Accuracy Precision Recall K-fold AUC-ROC Execution time | The result showed that oversampling followed by undersampling performs well for ensemble classification models. AIIKNN, SMTN, and RUS are performing well. SVM and KNN achieved perfect results. Best precision provided by oversampling followed by undersampling methods in conjunction with RF. NB classifier was the least. | Machine learning Group ULB. Kaggle. | NA |

| Article ID | ML/DL technique | Performance metrics | Results and value | Dataset | Future work |
|---|---|---|---|---|---|
| A165 | RF, SVM, ANN. | Accuracy | The result showed that ANN produced high accuracy then RF then SVM. | NA | Using more techniques. |
| A166 | KNN, isolation forest, local outlier factor | Accuracy Recall score | Results showed that all algorithms achieved 95.0% accuracy. Isolation forest had high accuracy and K-means produced the low accuracy. LR and vanilla LR gave great accuracy. | Real dataset/ Europeans cardholders | Implement an autoencoder or SVM. |
| A167 | LIGHTGBM AdaBoost, RF | Accuracy, precision and recall | The results showed that AdaBoost provided the highest result with 0.9613. In term of precision, Light BGM produces the highest result with 0.986. AdaBoost provided the highest recall with 0.889. | Real dataset/ Europeans cardholders/Kaggle | Adding more parameters. |
| A168 | OLightGBM RFLR SVM, DT, KNN NB | Accuracy Recall Precision F1-measure | The results highlight the importance of adopting an efficient parameter optimization strategy for enhancing the predictive performance. The proposed model outperformed other techniques with accuracy 98.40%. AUC is 92.88%. Precision is 97.34%. F1-score is 56.95%. | Real dataset/ Europeans cardholders/Kaggle | NA |
| A169 | RF, Apache Kafka | True positive rate (TPR), TNR, recall, precision accuracy | Using Apache Kafka to consume the transactions from the transaction record and publish them in real time. This project is using Cassandra as the storage layer. This proposed system offers the user maximum security and precision. | Data from the file system to the Cassandra database. | NA |
| A170 | Autoencoder RBM Federate learning | Accuracy ROC, Recall Precision | The results showed that the average accuracy of Autoencoder is 94% and RBM is 88%. AUC achieved a result of 0.94. | Real dataset/ Europeans cardholders | NA |
| A171 | RF | Accuracy Recall, Precision F1-score | The result showed that RF obtains good performance on small dataset. Some problems with imbalanced dataset. RF accuracy is 0.9632. Precision is 0.894. Recall is 0.85. F1-score is 0.871. | Real dataset/ Europeans cardholders | Improve RF itself |
| A172 | RF, LR, DT, KNN NB, Undersampling and oversampling techniques. | Accuracy Sensitivity Specificity Precision Matthews's co-relation | Results showed that LR is the best algorithm. The proposed classifier NN and LR outperform DT. LR accuracy is 0.9699. | Real dataset/ Europeans cardholders/Kaggle | NA |
| A173 | Local outlier factor, LOF, INFLO, and AVF | Accuracy Recall Precision | The results showed that using LOF, INFLO, and AFV resulted in the highest level of LOF. 96% accuracy, 98% recall, and 93% precision. | World website | Trying other algorithms. |

(Continued)

| Article ID | ML/DL technique | Performance metrics | Results and value | Dataset | Future work |
|---|---|---|---|---|---|
| A174 | LR, DT, SVM, NB, RF, KNN | Accuracy Precision Recall | The result showed that using RF obtained best accuracy of 99.947%, precision is 76%, and recall is 92.68%. | Real dataset/ Europeans cardholders | ANN can be used to construct new classification techniques. |
| A175 | Deep learning based fraud detection model (DLFD) | Accuracy Precision Recall | DL model is constructed for the prediction process using Keras. Comparison with existing models indicate high performance in detecting fraud. Detection rate is 8.7%. DLFD accuracy/0.997. Precision/0.929. Recall/0.795. | BankSim dataset was used for analysis of performance. | Improving the TPR levels and also on handling the concept drift. |
| A176 | ANN | Accuracy | The result showed that ANN is successful in fraud detection. Accuracy is 98%. However, ANN faced problems when training on huge datasets. | Dataset from company/South Africa | NA |
| A177 | ANN, GA, LR, SMOTE | Accuracy Precision Recall, F1-score | The results showed that the ANN with genetic algorithm obtained accurate results. The accuracy is 99.83%. Precision is 50.70%. Recall is 97.27%. F1-score is 66.66%. | Real dataset/ Europeans cardholders | NA |
| A178 | SVM, fuzzy association rules (FAR). Gradient recurrent unit | NA | The results showed that the proposed framework provided significant contribution. The framework allow to detect abnormal transaction. | NA | Implementation and evaluation the framework. |
| A179 | Hybrid ensemble-based. Boosting and bagging, RF, LR | MCC, Precision Recall Detection rate Accuracy | Results showed that the model is efficient in detecting fraud. MCC is 1.00. The false positive rate is 0.00235. False negative rate is 0.0003048. The detection rate is 0.9918. Accuracy is 0.9996. MCC is 0.9959. | Brazilian bank data and UCSD-FICO data | NA |
| A180 | Particle swarm optimization (PSO). NN | Accuracy Precision Recall | Results showed that performance of PSO is very high with 99.9% accuracy. | Real dataset/ European cardholders | Focus on solving imbalanced. |
| A181 | LR, RF Under sampling and oversampling | Confusion matrix, precision, F1-score, Roc-AUC | RF precision is 0.93. F1-score is 0.85. The oversampling, under sampling of data for accuracy of classifiers is promising. Oversampling technique gave better fraud prediction results as compared to random under sampling. | Real dataset/ European cardholders | NN and using combination of HMM or KNN to achieve better in fraud detection. |

### Funding

The authors received no funding for this work.

### Competing Interests

The authors declare that they have no competing interests.

### Author Contributions

- Eyad Abdel Latif Marazqah Btoush conceived and designed the experiments, performed the experiments, analyzed the data, prepared figures and/or tables, authored or reviewed drafts of the article, and approved the final draft.
- Xujuan Zhou conceived and designed the experiments, prepared figures and/or tables, authored or reviewed drafts of the article, and approved the final draft.
- Raj Gururajan conceived and designed the experiments, performed the experiments, prepared figures and/or tables, authored or reviewed drafts of the article, and approved the final draft.
- Ka Ching Chan conceived and designed the experiments, analyzed the data, prepared figures and/or tables, authored or reviewed drafts of the article, and approved the final draft.
- Rohan Genrich conceived and designed the experiments, prepared figures and/or tables, authored or reviewed drafts of the article, and approved the final draft.
- Prema Sankaran conceived and designed the experiments, prepared figures and/or tables, authored or reviewed drafts of the article, and approved the final draft.

### Data Availability

This is a literature review.

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
