# Peer review of "A systematic review of literature on credit card cyber fraud detection using machine and deep learning"

_PeerJ Computer Science, doi:10.7717/peerj-cs.1278_

## Round 0.1 · original submission · Minor Revisions

The review process is finished. The authors should revise the manuscript accordingly, as well as provide point-to-point responses to reviewers' concerns.

Reviewer 1 ·

Basic reporting

the introduction needs more additions, the introduction must be clearly visible and not only present the problem but provide a debate as to why this review study is needed

Experimental design

sound

Validity of the findings

sound

Additional comments

this review is very good and covers all aspects of the review. the results obtained also succeeded in answering the research questions given. but in the title section it is better to make a systematic review of literature rather than writing it as a comprehensive review

Reviewer 2 ·

Basic reporting

This paper represents a review article on the credit card cyber fraud detection. The review focuses specifically on exploring Machine Learning / Deep Learning approaches. By exploring six most well-known databases, the authors identified 181 research articles, published in three years (2019, 2020, and 2021).
The article is well written and use clear, unambiguous, and technically correct text.
The structure of the article is in accordance with a standard format for review papers: Introduction, Survey methodology, Research questions, Result and analysis, Trend of research, Gap analysis and the future direction, Limitation of the review, and Conclusions.
Figures are relevant to the content of the article, appropriately described and labelled.
The suggestion for the authors:
In the text it is stated in several places: “Our review selected papers that were published within the last three years, namely 2019, 2020, and 2021”. If accepted, the article will probably be published in 2023, which means that the mentioned years are not “last”. This issue can be solved by adding the research on papers from 2022; however, if this would be too demanding for the authors to solve in a reasonable time, then the word “last” should be removed from the text.

Experimental design

Research questions are well defined, relevant and meaningful. It is explained how the research fills an identified knowledge gap. The research is conducted in conformity with the prevailing ethical standards in the field.

Validity of the findings

The obtained results are of interest to the general and academic audience. The conclusions are appropriately stated and connected to the original investigated questions.

Reviewer 3 ·

Basic reporting

The title of the paper is in correspondence with its content.
Also, the summary and conclusion correspond to the essence of the work.
The acronyms are consistently explained when first used in the text.
The use of terminology is correct and it complies with applicable standards.
I think that the labels of Figure 3, Figure 4 and Figure 5 name should be different (without shows) and also Table 7 is wrongly referenced twice in the text (Appendix C, not B). Also, the title of Figure 5 is wrong ("Number of articales")

Experimental design

The review paper comprehensively covers the latest research (2019-2021). Can the authors briefly explain the motive for singling out this period or statistically present the number of works in the past ten years in the Introductory part?
The authors clearly explained the methodology and method of literature selection and presented it clearly in the text, graphically, and tabularly.
Several forms of cyber fraud are given in the Introduction: actual credit card theft, theft of confidential credit card information, and credit card information being entered without the cardholder's permission during an online transaction. Are statistical data available to the authors on the share of these different forms of fraud?

Validity of the findings

In the paper, the authors clearly identified the achieved goal of the research and, according to the defined research questions, provided guidelines for further work in this area
The consistency of the obtained results and comparability indicate the global importance of the research, with special reference to the number of research papers by country (Figure 5).

Additional comments

No comment

---

## Round 0.2 · accepted · Accept

The review paper is accepted.

Reviewer 1 ·

Basic reporting

Sound

Experimental design

Sound

Validity of the findings

Sound

Additional comments

Ok, Accept

Reviewer 2 ·

Basic reporting

The authors have satisfactorily addressed the reviewers comments.

Experimental design

The authors have satisfactorily addressed the reviewers comments.

Validity of the findings

The authors have satisfactorily addressed the reviewers comments.

Reviewer 3 ·

Basic reporting

no comment

Experimental design

no comment

Validity of the findings

no comment

Additional comments

I think that the authors have very conscientiously and precisely responded to the comments and suggestions of the reviewers and thereby improved the quality of this important review paper, which can represent a significant starting point for future research in this area.

Reviewer 4 ·

Basic reporting

The paper is interesting and deals with important tasks. Review papers are not often written, but good review papers can be useful for the wider community.

Experimental design

The paper has a good structure with all the necessary elements. The authors have clearly shown problem formulation.

Validity of the findings

This part of the paper is well presented with enough Tables and Figures.

Additional comments

The paper falls within the scope of the journal PeerJ computer science. I must note that I am not be reviewer of this manuscript in the first round. Now in the second round I have assessed the paper in the whole and replies to other reviewers.
My decision is that paper can be accepted and forward to production. The authors well elaborated problem and made really good review paper. Number of elaborated studies is enough and number of total references is 192 which is proof of time and effort needed for writing such study.
The abstract is well written, while keywords are selected in the proper way.
Generally the paper has potential and can be cited in the future, especially because research interest of this study is very popular.